



# The influence of accretionary orogenesis on subsequent rift dynamics

Zoltán Erdős[1,2], Susanne J. H. Buiter[1,2], Joya L. Tetreault[3]

[1]GFZ Helmholtz Centre for Geosciences, Potsdam, Germany
[2]RWTH Aachen University, Tectonics and Geodynamics, Aachen, Germany
[3]Geological Survey of Norway (NGU), Trondheim, Norway

*Correspondence to*: Zoltán Erdős: erdoes@gfz.de

**Abstract.** The Wilson Cycle of closing and opening of oceans is often schematically portrayed with 'empty' oceanic basins. However, bathymetric and geophysical observations outline anomalous topographic features on the ocean floor, such as

microcontinents and oceanic plateaus, that can be accreted or subducted when oceans close in subduction. If later rifting and extension localizes in the area of former oceanic closure, this implies that the rifted margins formed in regions characterized not only by continent-continent collision, but also by the presence of accreted continental terranes. An excellent example of such a system can be found in the North Atlantic, where the late-Paleozoic to Mesozoic opening of the Atlantic Ocean occurred immediately after the early Paleozoic Caledonian orogeny, that formed during the collision of Baltica and Laurentia continents

but also incorporated allochthonous continental terranes. The full evolution from subduction to accretion-collision and how those processes bear on continental rifting has not been studied systematically. Potential factors that can influence the evolution and structural style of a rift in such a tectonic setting include the thermo-tectonic age of the orogen, the number and type (size, rheology) of accreted terranes, the nature of terrane boundaries, as well as the velocity of rifting.

Here, we use 2D finite-element thermo-mechanical models to investigate how accreted microcontinents and the size of the

orogen affect the style of continental rifting. Our models demonstrate that there is a competition between thermal and structural inheritance that has a first order effect on the style of rifting. In large, warm orogens thermal inheritance dominates over structural inheritance, leading to the formation of new major extensional shear zones, whereas in small, cold orogens structural inheritance dominates over thermal inheritance, allowing for efficient deformation localization along pre-existing sutures. In comparison, the presence of accreted terranes within the orogen only has secondary effects. In small, cold orogens, when

multiple sutures are present, the oldest, shortest and most optimally oriented suture is reactivated extensively, with the others experiencing only limited activity. In contrast, in large, warm orogens, the suture closest to the centre of the orogen is inverted the most. Additionally, the presence of accreted terranes within the pre-rift lithosphere leads to the formation of continental fragments in the rifted margin architecture.

**Plain language summary**

We used computer models to study how mountains formed by the collision of tectonic plates can later affect the breakup of these same plates. Our results show that in large, warm mountain belts, new faults form due to the orogen being overall



weak, while in smaller, colder belts, breakup follows old fault zones. Microcontinents that were accreted during collision can create new continental fragments during extension. These findings help explain how past geological events shape continent margins.

## 1. Introduction

### 1.1. Types of inheritance

The Wilson Cycle that describes the closing of oceans and opening of new ocean basins along their inherited sutures has proven a powerful concept in analysing the deformational histories of plate margins (Wilson, 1966). In its initial formulation, it described a process of oceanic subduction, continental collision, continental extension and formation of a new ocean for the
North Atlantic Region. Since its inception, it has been recognized that most present-day rifted continental margins were formed along former collision zones (Buiter and Torsvik, 2014) and that many collisional orogens were built from rifted margins (e.g., Jackson, 1980; Manatschal et al., 2021; Tavani et al., 2021). This implies that continental collisional and extensional processes will usually involve features inherited from previous phases of deformation.

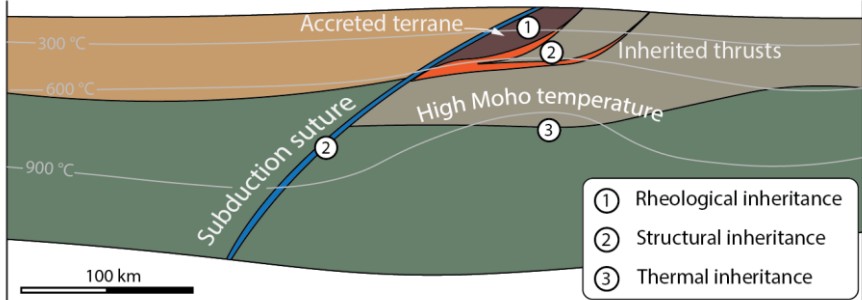

**Figure 1. A simplified lithospheric-scale cross-section illustrating the different types of orogenic inheritance that can exert a controlling influence on a subsequent phase of continental rifting.**

Orogenic inheritance can span a wide range of features, from inherited weak thrust faults, specific sedimentary and magmatic units, long-term lithospheric thermal perturbation, to amalgamation of different lithological terranes and crustal thickness variations (Manatschal et al., 2015). Manatschal et al. (2015) classified these inherited features into three groups: 1)
rheological, 2) structural, and 3) thermal inheritances and we use this terminology here (Fig. 1). In most cases the different types of inheritances appear in combination, making it difficult to determine their individual contributions. Sutures, representing structurally weak zones, can emplace lithospheric blocks with very different rheological characteristics, representing rheological inheritance. Thickened crust due to a previous phase of shortening can also represent a rheological inheritance, but – with enough time afforded – the increased thickness of crustal rocks that are rich in heat-producing elements
can result in a high geothermal gradient and thus in thermal inheritance. Since later stages of rifting and rifted margin evolution





tend to overprint earlier deformation phases, several well described examples for these inheritance types come from the East African Rift System, where continental rifting is still active.

An example of plate-scale rheological inheritance is the relatively rigid Archean Tanzanian craton south of the Main Ethiopian
Rift, where the Eastern and Western Branches of the East African Rift System wrap around the cratonic block (Fig. 2), deforming Proterozoic orogenic belts that formed during the amalgamation of the African Continent (e.g., Corti et al., 2007; Daly et al., 1989; McConnell, 1967; Samsu et al., 2023) and references therein). Geophysical investigations in the Main Ethiopian Rift also point to the strong role of structural inheritance (Corti, 2009). The combined observations of a string of studies suggest that rift location and initial evolution was probably controlled by a NE–SW-trending lithospheric-scale pre-
existing heterogeneity corresponding to a suture zone separating two distinct Proterozoic basement terranes (Bastow et al., 2008; Bastow et al., 2005; Keranen and Klemperer, 2008; Keranen et al., 2009).

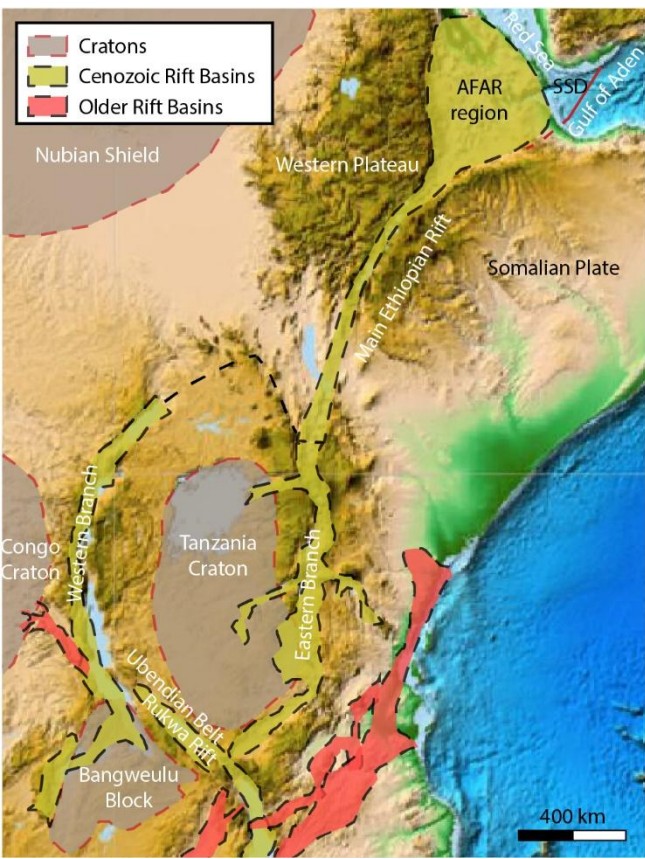

**Figure 2. Map of the rheological, structural and thermal inheritance example sights from the East African Rift System. The shapes of cratons and rift basins were drawn after Ali and Watts (2013); Kidane et al. (2003); Kolawole et al. (2021); Rime et al. (2023);**
**Samsu et al. (2023)**

The original Wilson Cycle formulation focused on the recognition of earlier oceans preceding continental collision and subsequent ocean formation and was thus, for simplicity, described as an "empty" ocean basin. It has, however, long been



recognized that oceans feature multiple bathymetric highs, such as microcontinents, island arcs, seamounts, or oceanic plateaus. These features can accrete to the overriding continent during subduction and in a subsequent phase of continent-continent collision, they can become part of the orogen. The accretion of terranes adds multiple subduction sutures to the orogen through trench jumps, thus adding structural inherited features that can affect later deformation phases.

An example of such structural inheritance, where a previous accretionary suture was later reactivated during rifting as a major border fault is found in the Rukwa Rift in East Africa, where the Precambrian Chisi Shear Zone and the adjacent terrane boundaries represent zones of mechanical weakness in the pre-rift basement. The Ubendian Belt, formed during the collision of the Archean Tanzania Craton and the Bangweulu Block, which comprises several distinct terranes (Boniface and Schenk, 2012; Daly, 1988; Daly et al., 1989; Lenoir et al., 1994). This weakness-zone controlled the first-order strain distribution and rift development during the earliest phase of extension (Kolawole et al., 2021; Manighetti et al., 2001).

Natural examples of thermal inheritance influencing rifting are far more difficult to identify, as they are transient features by nature. Advection of cold material in a subduction zone to great depths can result in an anomalously cold lithosphere. However, thickened orogenic crust can result in increased radiogenic heat production that over time can produce a warm lithosphere. The picture is further complicated by erosion that can remove heat-producing material cooling the lithosphere, but can also result in the rapid exhumation of hot material to near surface depths. In the East African Rift System, the preceding orogeny is Precambrian in age and could thus be expected to be either warm and weak, because of increased heat production in its thickened crust, or cold and stiff, if erosion removed part of the crust. The presence of the African hotspot (Manighetti et al., 2001; Rime et al., 2023) adds further complexity. It has been proposed that the locally high heat-flux above the hotspot weakened the lithosphere and the diffuse deformation within this weak block inhibited localization of deformation adjacent to the Shukra-el-Sheik discontinuity (Kidane et al., 2003; Manighetti et al., 2001; Manighetti et al., 1997), that separates the nonvolcanic central and eastern segments of the Gulf of Aden from the plume-influenced volcanic western segment (Ali and Watts, 2013). This process has stalled the propagation of rifting from the Gulf of Aden for at least 7 Myrs, thereby altering the rift-propagation dynamics (Kidane et al., 2003; Manighetti et al., 2001; Manighetti et al., 1997). This illustrates how locally high heat flow can override the effects of structural inheritance (Kidane et al., 2003).

### 1.2.    Previous numerical model studies investigating the role of orogenic inheritance in rifting

The studies of Butler et al. (2015), Petersen and Schiffer (2016), Salazar-Mora et al. (2018), Chenin et al. (2019), Peron-Pinvidic et al. (2022) and Salazar-Mora and Sacek (2023) used numerical experiments to explore the effects of orogenic inheritance on rifting. Butler et al. (2015) tailored their model setup to the Norwegian rifted margin and explored the effect of **rheological inheritance** and melt-weakening during orogeny and post-orogenic quiescence on subsequent rifting. Their models also included multiple terranes sandwiched between two continents and found that depending on the strength of the enclosing continental lithosphere, units can experience different degrees of burial and exhumation. Petersen and Schiffer (2016) compared rifting of a laterally homogeneous continent to a continent which included a simplified suture and a hydrated wedge above, to explore the effects of **structural inheritance** on rift formation. In their models, the suture is reactivated as a



major extensional detachment, but as the suture was only defined within the mantle-lithosphere, it did not rupture the crust, where instead a set of newly formed conjugate normal faults were formed. Salazar-Mora et al. (2018) found that the **rheology** of the continental lithosphere (controlling the strength of the upper crustal shear zones and the degree of decoupling of upper crust from lower crust) and the size of the orogen have a first order effect on the structure of the resulting rifted margins.

Subsequently, Salazar-Mora and Sacek (2023) explored the effects of tectonic quiescence between orogeny and rifting – influencing **thermal inheritance** – and found that the length of the quiescent period is important for cases in which convergence formed a large orogen. The conductive warming of the thickened lithosphere during the quiescent period allows for the development of a broad thermal weak zone, leading to a more distributed deformation style during rifting. Chenin et al. (2019) explored the effects of wide-spread, thick mafic underplating at lower-crustal level below inherited crustal structures

of an older orogen on a subsequent phase of rifting. This setup incorporates both **structural** and **rheological inheritance** to simulates observations of the setting below the Variscan orogenic structures of Western Europe, where rifting during the opening of the Tethyan and the North Atlantic did not localize at former suture zones. Chenin et al. (2019) found that such underplating located in thermally equilibrated lithosphere can suppress reactivation of crustal scale inherited structures. Peron-Pinvidic et al. (2022) used a detailed, multiphase rifting history, tailored to the opening of the North Atlantic Ocean, to study

the effects of **structural inheritance** on reactivation patterns. They show that even though a model without orogenic inheritance can reproduce the main characteristics of rifted margins, including distinct structural domains as consistently observed worldwide, the architectures generated by a numerical simulation that include a phase of pre-rift orogeny are more complex, with a high degree of geometrical details that are similar to many observations made on natural cases.

Here we take these previous studies a step further by geodynamically modelling an entire Wilson-cycle – bar subduction
initiation – from the closure of an oceanic domain through subduction and continent-continent collision to rifting and continental breakup. In addition, several of our experiments also include accretion of 1 or 2 microcontinents, effectively varying the number of major inherited structures in the collisional orogen. We aim to explore the effects of rheological, structural and thermal orogenic inheritance on continental rift evolution and first order continental margin-architecture. Our setup is not aimed at reproducing the evolution of a specific rift system on Earth. Our goal is rather to explore the underlying

physical processes in a way that is transferrable between systems. Understanding the potential impact of rheological, structural and thermal inheritance on the activity of subsequent fault networks has implications for seismic hazards assessment (e.g., Fonseca, 1988; Hecker et al., 2021; Wedmore et al., 2020), geothermal energy (e.g., Bertrand et al., 2018; Yeomans et al., 2021), mineral exploration (e.g., Glerum et al., 2024; Rowland and Sibson, 2004), including the potential for natural hydrogen reservoirs (e.g., Vasey et al., 2024; Zwaan et al., 2025) and $CO_2$ and nuclear waste storage (e.g., Andrés et al., 2016).





## 2.  Experimental setup

We perform 2D numerical experiments on a Cartesian region that is 660 km deep and 2000 km wide for the zero-microcontinent model, 2300 km wide for the one-microcontinent model and 2800 km wide for the two-microcontinent model (Fig. 3).

| | Unit | Syntect. Sedim. | Continental crust | | Microcontinent | | | Oceanic crust | | | | Lith. Mantle | Asthenosphere | |
|---|---|---|---|---|---|---|---|---|---|---|---|---|---|---|
| | | | Upper crust | Lower crust | Upper crust | Middle crust | Lower crust | Sedim. cover | Crust | Eclogite sedim. | Eclogite crust | | | |
| **Mechanical parameters** | | | | | | | | | | | | | | |
| $z$ | km | - | 20 | 20 | 8 | 10 | 7 | 2 | 5 | - | - | 80 (73) | - | |
| $\rho_0$ | kg m$^{-3}$ | 2650 | 2760 | 2900 | 2700 | 2850 | 3000 | 2900 | 2900 | 3365 | 3370 | 3370 | 3370 | |
| $T_0$ | °C | 0 | 0 | 0 | 0 | 0 | 0 | 0 | 0 | 0 | 0 | 600 | 600 | |
| $\phi$ | deg | 2° | 20°-10° | 20°-10° | 20°-10° | 20°-10° | 2° | 2° | 5°-2° | 2° | 5°-2° | 20°-10° | 20°-10° | |
| $C$ | Pa | 5 10$^6$ | 2 10$^7$ | 2 10$^7$ | 2 10$^7$ | 2 10$^7$ | 2 10$^6$ | 5 10$^6$ | 1 10$^7$ | 5 10$^6$ | 1 10$^7$ | 2 10$^7$ | 2 10$^7$ | |
| fl | - | WQ[1] | WQ[1] | WA[2] | WQ[1] | WA[2] | G[4] | WQ[1] | G[4] | WQ[1] | G[4] | WO[3] | WO[3] disl. | WO[3] diff. |
| $f$ | - | 1 | 1 | 1 | 1 | 1 | 1 | 1 | 1 | 1 | 1 | 5 | 1 | 1 |
| $A$ | Pa$^{-n}$ s$^{-1}$ | 8.57 10$^{-28}$ | 8.57 10$^{-28}$ | 7.13 10$^{-18}$ | 8.57 10$^{-28}$ | 7.13 10$^{-18}$ | 1.12 10$^{-10}$ | 8.57 10$^{-28}$ | 1.12 10$^{-10}$ | 8.57 10$^{-28}$ | 1.12 10$^{-10}$ | 1.76 10$^{-14}$ | 1.76 10$^{-14}$ | 1.76 10$^{-26}$ |
| $Q$ | J mol$^{-1}$ | 2.23 10$^5$ | 2.23 10$^5$ | 3.45 10$^5$ | 2.23 10$^5$ | 3.45 10$^5$ | 4.97 10$^5$ | 2.23 10$^5$ | 4.97 10$^5$ | 2.23 10$^5$ | 4.97 10$^5$ | 4.3 10$^5$ | 4.3 10$^5$ | 2.4 10$^5$ |
| $n$ | - | 4 | 4 | 3 | 4 | 3 | 3.4 | 4 | 3.4 | 4 | 3.4 | 3 | 3 | 1 |
| $V$ | m$^3$ mol$^{-1}$ | 0 | 0 | 3.8 10$^{-5}$ | 0 | 3.8 10$^{-5}$ | 0 | 0 | 0 | 0 | 0 | 1.5 10$^{-5}$ | 1.5 10$^{-5}$ | 0.5 10$^{-5}$ |
| $d$ | m | - | - | - | - | - | - | - | - | - | - | - | - | 1 |
| $m$ | - | - | - | - | - | - | - | - | - | - | - | - | - | 2.5 |
| $w$ | - | - | - | 1 10$^3$ | - | 1 10$^3$ | - | - | - | - | - | - | - | - |
| $r$ | - | - | - | 1 | - | 1 | - | - | - | - | - | - | - | - |
| **Thermal parameters** | | | | | | | | | | | | | | |
| $c_p$ | m$^2$K$^{-1}$s$^{-2}$ | 1000 | 900 | 800 | 900 | 800 | 750 | 750 | 750 | 750 | 750 | 750 | 750 | |
| $k$ | W m$^{-1}$K$^{-1}$ | 2.5 | 2.5 | 2.5 | 2.5 | 2.5 | 2.5 | 2.5 | 2.5 | 2.5 | 2.5 | 2.5 | 87.6 | |
| $\alpha$ | K$^{-1}$ | 0 | 0 | 0 | 0 | 0 | 0 | 0 | 0 | 0 | 0 | 2.5 10$^{-5}$ | 2.5 10$^{-5}$ | |
| $H$ | µW m$^{-3}$ | 1 10$^{-6}$ | 0.95 10$^{-6}$ | 0.7 10$^{-6}$ | 0.95 10$^{-6}$ | 0.7 10$^{-6}$ | 0 | 0 | 0 | 0 | 0 | 0 | 0 | |
| $\eta_{eff}$ | Pa s | 2 10$^{19}$-10$^{26}$ | | | | | | | | | | | | |

**Table 1 Mechanical and thermal material properties used in the experiments, where z is Thickness, $\rho_0$ is Reference density, $T_0$ is Reference temperature, φ is Angle of friction, C is Cohesion, fl is Flow-law, f is Scaling factor, A is Pre-exponent factor, Q is Activation energy, n is Power-law exponent, V is Activation volume, d is Grain size, m is Grain-size exponent, w is Water content, r is Water content exponent, $c_p$ is Heat capacity, k is Heat conductivity, α is Thermal expansion, H is heat productivity and $\eta_{eff}$ is Viscosity range. Value in parentheses: Thickness value mantle lithosphere for the oceanic plate. [1]Gleason and Tullis [1995]; [2]Rybacki et al. [2006]; [3]Karato and Wu [1993]; [4]Wilks and Carter [1990]**

We use the thermo-mechanical finite-element code SULEC v.4 (Ellis et al., 2011; Naliboff and Buiter, 2015; Naliboff et al., 2017; Tetreault and Buiter, 2012, 2018) for which a detailed description is provided in Appendix A1. The experimental setup



builds on the study of (Erdős et al., 2024) (under revision), in which we examined oceanic subduction and continental collision in the presence of 0, 1 or 2 microcontinents. Here, we extend a subset of these experiments to investigate first a phase of
tectonic quiescence and thermal relaxation and finally a phase of extension, driving continental rifting.

The initial layer-cake rheology (for the exact material properties see table 1) has an adiabatic sublithospheric mantle at the bottom, characterized by parallel diffusion and dislocation creep of wet olivine (Karato and Wu, 1993). Above this layer, on the left side of the model, lies the overriding continental plate, which consists of an 80 km-thick mantle lithosphere (wet olivine dislocation creep with a scaling factor f = 5), a 20 km-thick lower crust (wet anorthite dislocation creep; Rybacki et al., 2006),
and a 20 km-thick upper crust (wet quartz; Gleason and Tullis, 1995). To the right of the overriding plate is an oceanic domain, comprising a 73 km-thick mantle lithosphere (wet olivine dislocation creep with a scaling factor f = 5), a 5 km thick oceanic crust (gabbro dislocation creep; Wilks and Carter, 1990), and a 2 km-thick sedimentary cover (gabbro with lowered cohesion and internal angle of friction). These two domains are separated by a "compositionally weak seed" representing a proto-subduction zone with a dip of 30° that helps deformation localization during experiment initiation (Tetreault and Buiter, 2012).
This weak seed consists of oceanic crustal material overlain by a layer of oceanic sediments, underplating a sloped continental margin. To the right of the oceanic domain, the trailing continent replicates the structure of the overriding plate. A steep (60°) continental margin separates the trailing continent from the ocean.

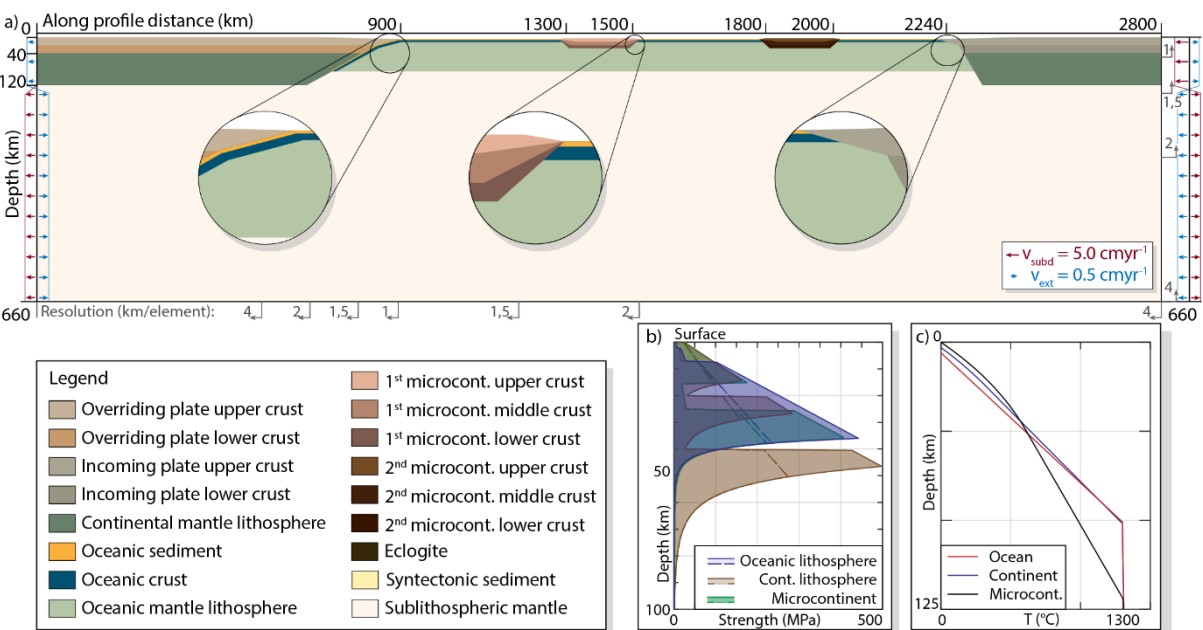

**Figure 3. Initial experimental setup before the initiation of convergence, shortening, terrane accretion and orogeny. a) The initial**
**geometry of a model containing two microcontinents with close-ups of the ocean-continent margins and an ocean-microcontinent margin. The black numbers along the vertical domain edges represent distance in kms measured from the top left corner. The light-gray numbers along the edges represent the numerical resolution, measured as element size in kms. The red and blue arrows along the sides represent the velocity boundary conditions in the collision and rifting phases respectively. b) Analytically-calculated strength-profiles of the initial continental lithosphere, the oceanic lithosphere and a microcontinent, with dashed lines showing the**
**strain-weakened profiles. c) Analytical temperature profiles of the oceanic and continental lithospheres as well as the microcontinents used in the strength calculations depicted in b).**





Models **M0a-c** contains an "empty" oceanic plate, while models **M1a-b** and **M2a-b** contain one and two microcontinents respectively, embedded within the oceanic lithosphere. The microcontinents rise 2.5 km above the oceanic seafloor (that is, in turn, 5 km below the surface of the continental blocks) and consist of three layers: a 10 km-thick gabbroic lower crust, a 7 km-thick wet anorthite mid-crust and an 8 km-thick wet quartz upper crust. The oceanic basin between the trench and the first terrane is 400 km wide, each microcontinent is 200 km wide and the two microcontinents are separated by a 300 km wide basin. The trailing continent is 250 km away from edge of the second terrane. The numerical resolution of the model varies along the profile and with depth; with the highest resolution of 1 km by 1 km around the subduction zone (for the details see grey numbers along the axes of Fig. 3).

The temperature is fixed at 0 °C at the surface of the model and 1435 °C at the base of the model, while the side-boundaries are thermally insulated. Using the thermal parameters assigned to the materials we initialize the temperature field with a steady-state solve before allowing the field to evolve during the experiments. In order to mimic active mantle convection at a high Nusselt number, the thermal conductivity of the sublithospheric mantle is increased stepwise from 2.5 to 87,6 $Wm^{-1}K^{-1}$ when above a threshold of 1,300 °C. This approach has been widely used in subduction as well as rift modelling studies (Butler and Beaumont, 2017; Pysklywec and Beaumont, 2004; Tetreault and Buiter, 2012; Warren et al., 2008) as it prevents secular cooling of the model domain while maintaining a constant vertical heat flux at the base of the lithosphere and keeping the mantle close to the adiabatic gradient.

To prevent the subducting slab from interacting with the base of the model, the slab materials are arbitrarily transformed into sub-lithospheric mantle when reaching approximately 625 km depth (at 200 GPa pressure and 900 °C temperature). This simplistic approach somewhat limits the slab-pull force but the present setup still reaches values up to $2.5 \cdot 10^{13}$ $Nm^{-1}$ (comparable to values calculated for other similar studies e.g., Erdős et al., 2021; Wolf and Huismans, 2019), while keeping the computational costs of running these experiments manageable by allowing for a shallower computational domain.

The kinematic boundary conditions for our experiments consist of a free surface, free slip on the walls and base of the model and fixed in- and outflow velocities at the continental lithosphere walls that are equivalently balanced with velocities on both walls across the sub-lithospheric mantle (Fig. 3). All experiments consist of three phases: **1)** an initial phase of convergence (*collision phase*) characterized by a 5 cm $yr^{-1}$ inflow velocity; **2)** followed by a phase of thermal relaxation with no in- or outflow through the model boundaries (*relaxation phase*); **3)** before a final phase of extension (*rifting phase*) where 0.5 cm $yr^{-1}$ outflow velocities are prescribed at both walls within the lithosphere (see red and blue arrows on Fig. 3). Some of the experiments exhibited a phase of late-stage hyper-extension that occurs through rift migration at an almost constant velocity, accomplished by sequential, oceanward-younging, upper crustal normal faults. This upper-crustal normal faulting is accompanied by a lower crustal flow towards the ocean. This agrees with Brune et al. (2016), who argue that a rift-intrinsic strength-velocity feedback causes an increase in extension velocity late in the extension process. Such changes in rift velocity can be achieved by switching from a constant velocity to a constant stress boundary condition. We here approximate this behaviour by identifying the time in model evolution when deformation moves into the hyper-extension regime, at which time,



we increase the extension velocity five-fold, achieving swift crustal breakup. This is applied in experiments **0Ma**, **0Mb**, **1Ma**
and **2Ma**.

The length of the collision phase varies between the different models in order to achieve different orogen sizes, but the
relaxation phase is always run until the 130 Myr experimental time. This means that the length of the relaxation phase varies
between the model experiments, but it was chosen such as to allow for slab breakoff and reaching a nearly steady-state thermal
structure. The rifting phase is run until full crustal breakup is achieved.

### 2.1.     Limitations of the experimental setup

Models are inherently simplified representations of nature, so it is essential to consider their limitations. The experimental
setup is **two-dimensional**, ignoring possible along-strike variations in orogenic structure that could influence rift evolution.
While the effects of a 3D slab and finite-width microcontinents in the collisional phase are discussed in (Erdős et al., 2024; in
review), here we focus on their potential impact on the ensuing thermal relaxation and rifting. In nature, slab breakoff often
propagates as a tear along the subduction zone (Burkett and Billen, 2010; Rosenbaum et al., 2008; van Hunen and Allen, 2011;
Yoshioka and Wortel, 1995), affecting detachment depth and orogen geometry before extension. Earlier slab breakoff could
accelerate thermal equilibration, rebound, erosion, and the removal of radiogenic heat-producing upper crust, potentially
initiating lower crustal flow sooner. During rifting, an extension direction oblique to the orogen's strike might further
complicate reactivation of collisional structures in extension. Additionally, inherited transform faults could significantly
influence deformation by vertically partitioning the lithosphere (Thomas, 2006). However, further exploration of these 3D
effects is beyond this study's scope.

Using a **fixed velocity boundary condition** results in variable far-field forces that can strongly influence model evolution.
The 0.5 cm yr$^{-1}$ outflow velocity at the side boundaries is within the typical range for rift systems (Lavier and Buck, 2002).
Generally, lower extension velocities yield more asymmetric rifting (Huismans et al., 2005) and may promote basin migration
(van Wijk and Cloetingh, 2002) while higher velocities favour heat advection, weakening the rift through asthenospheric
upwelling and lead to a continuous strength loss (Buiter et al., 2023). Recent studies also indicate that rift-intrinsic strength-
velocity feedback, which can be replicated using constant force boundary conditions, drives two-phase rift velocities (Brune
et al., 2016). Our models, using an intermediate extension velocity, produce wide rifted margins and hyper-extension,
suggesting that, in these experiments, plate velocity affects margin structure less than thermal and structural inheritance.

Strain weakening is included in the models, but no **strain-healing mechanism** is implemented. Strain healing has been
proposed to increase resistance in inherited structures over time since their last activity; however, the exact nature and
parameterization of these processes remain poorly understood. Including strain healing would likely increase the role of
thermal over structural inheritance. Nonetheless, natural examples, such as Proterozoic basement structures reactivated during
Cenozoic rifting, suggest that structural inheritance can be highly long-lasting (Ring, 2010).

Furthermore, **surface processes** are represented by a very weak surface diffusion algorithm, while it has been shown that they
can have an effect on the style of rifting (e.g., Neuharth et al., 2022 and references therein) as well as on other features, like





melt-production and volcanism (Sternai, 2020). They can further affect the thermal state of the lithosphere prior rifting through the efficient removal of the upper crust, that can be rich in heat producing elements.

Likewise, our models only include a **simplistic approximation of** the eclogite phase transition and neglect latent heat and all other **phase changes**. The inclusion of more realistic phase change rules could significantly alter the detachment and accretion of microcontinental material at the base of the overriding plate. It also means that compositional inheritance, as defined by Manatschal et al. (2015) appears only in a very limited capacity in the presented models, and mainly in correlation with structural inheritance.

Finally, **melt production and migration** are not accounted for in our experiments. In the collisional phase, these processes would primarily affect the rheology of the initial overriding plate (potentially lowering its integrated strength). During rifting, they could have a strong effect, especially for those experiments where the mantle-lithospheric root of the orogen is delaminated during the thermal relaxation phase. This would further reinforce the tendency of these experiments to localize deformation away from the inherited sutures, within the thick orogenic root.

**3.    Model Results**

**3.1.    Experiment series**

We present and analyse the results of seven numerical experiments. **M0a**, **M0b** and **M0c** are experiments without microcontinents, where an "empty" oceanic basin is subducted between two continents and the duration of continent-continent collision is varied. In experiment **M0a** the shortening phase is 13 Myr long; lasting until the oceanic basin is closed with an

approximately 40 km-thick lithosphere in the centre of the model (i.e., small orogen). In experiment **M0b**, the shortening phase lasts until 15.5 Myr. This time was chosen because the additional convergence allowed for an amount of material equivalent to one microcontinent (of the size used in this study) to enter the subduction zone (i.e., large orogen). Finally, in experiment **M0c**, the shortening phase lasts until 18 Myr allowing for an amount of material equivalent to two microcontinents to enter the subduction zone (i.e., extra-large orogen).

**M1a** and **M1b** are experiments with one microcontinent embedded within the oceanic domain. Similarly to the **M0** model series, the difference is in the duration of the shortening phase with **M1a** (18.6 Myr, only closure) generating a small orogen and **M1b** (21 Myr, an additional one microcontinent equivalent material in the subduction zone) resulting in a large orogen. Finally, using the same logic as above, **M2a** and **M2b** are experiments with two microcontinents embedded within the oceanic domain and the respective shortening phases last for 28 Myr (small orogen) and 30 Myr (large orogen).

The collision phase of each of these experiments has been presented in detail and analysed in (Erdős et al., 2024; in review) so we only provide their description in Appendix A2. Below we give a short summary of the behaviour of each model from the onset of thermal relaxation until crustal breakup and provide a detailed account of their temporal evolution in Appendix A3.



### 3.2.     Summary of model behaviours

In experiment **M0a** (Fig. 4a-d and Supplementary Animation 1; for viscosity plots see Fig. C2a-d) breakoff happens through the necking of the slab at 200 km depth leaving a long sliver of oceanic lithosphere sandwiched between the two collided continents, resulting in a cold lithosphere and a very small orogen. Prior to the onset of rifting, both structural and rheological inheritance are present through the presence of the suture and the preserved oceanic mantle-lithosphere respectively. The subsequent extension reactivates and inverts the suture creating a large-scale core-complex style rift with largely undeformed

continents on either side.

In experiment **M0b** (Fig. 4e-h and Supplementary Animation 2; for viscosity plots see Fig. C2e-h) the slab that hangs below the orogen is slowly wrenched off by its own weight with the detachment forming in the incoming continental portion of the orogenic lithosphere. The result is a dome-shaped asthenospheric upwelling in the lithosphere, under thickened crust (75 km), supporting a 4 km-high, 200 km-wide, hot (Moho temperatures locally reaching 900 °C) orogen. Prior to the onset of rifting

all three inheritance-types are present, as the suture is combined with a thickened crust and a high geothermal gradient. During the subsequent phase of rifting the suture is reactivated as a normal shear zone but the hot, thickened incoming continental crust is also heavily deformed, resulting in an asymmetric rift with one wide rifted margin.

In experiment **M0c** (Fig. 4i-l and Supplementary Animation 3; for viscosity plots see Fig. C2i-l) the slab breaks off at the base of the lithosphere relatively swiftly during the relaxation phase, followed by a long period of ductile deformation deep in the

crust. At the end of the relaxation phase a 250 km-wide orogen with 80—90 km thick, hot crust and a maximum elevation of 5 km sits in the footwall of the suture. Once again, prior to the onset of rifting all three inheritance-types are present, as the suture is combined with a heavily thickened crust and high geothermal gradient in a wide orogen. During the rifting phase, deformation primarily localizes in this orogen and largely away from the suture, creating two wide rifted margins while leaving the overriding plate largely undeformed. The rifted margin on the side of the overriding plate includes allochthonous material

(crust and mantle of the collided continent).



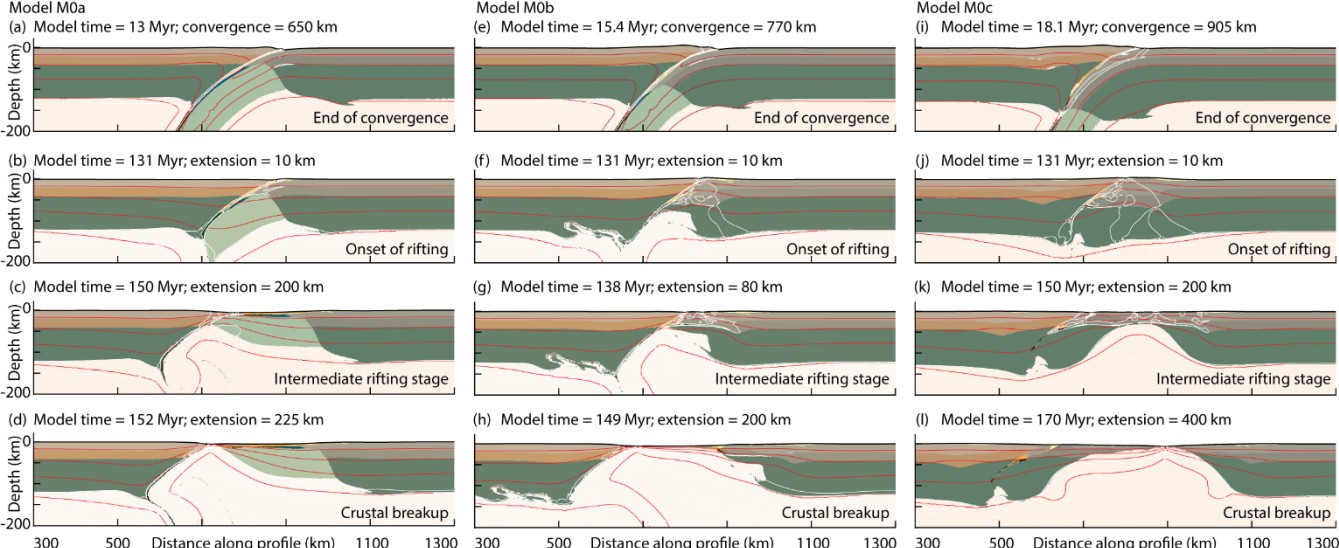

**Figure 4. Models M0a M0b and M0c with no microcontinents; (a-d) M0a characterized by a small orogen, exhibiting full inversional reactivation of the subduction suture during rifting. Extension is solely localized on inherited suture; (e–f) M0b characterized by intermediate size orogen, exhibiting full inversional reactivation of the subduction suture during rifting accompanied by extensive deformation away from the suture in the hot orogen; (i-l) M0c characterized by a large orogen, exhibiting partial inversional reactivation of the subduction suture and rifting away from the suture in the hot orogen. Material colours (see legend of Fig. 1) at key time steps, with selected isotherms displayed by red contours (300°C, 600°C, 900°C, and 1300°C) and areas experiencing high strain rates displayed with white contours ($10^{-13}$ to $10^{-15}$ s$^{-1}$).**

The evolution of the single microcontinent experiment **M1a** (Fig. 5a-d and Supplementary Animation 4; for viscosity plots see Fig. C3a-d) is very similar to that of experiment **M0a** with no microcontinents. During the relaxation phase the slab undergoes a very slow process of necking at around 200 km-depth, leaving a long sliver of oceanic lithosphere still attached to the accreted terrane, sandwiched between the two collided continents and resulting in a less than 200 km-wide and 1.5 km-high orogen. Similarly to **M0a**, both structural and rheological inheritance are present prior to the onset of rifting; the former through the presence of two sutures while the latter in the form of the accreted terrane and preserved oceanic mantle-lithosphere beneath it. During the rifting phase, the older subduction interface is extensively reactivated and inverted with most of the extensional deformation focused within the highly pre-deformed leading edge of the accreted terrane. The system acts as a giant core complex as the oceanic crust and mantle-lithosphere is slowly exhumed. Full crustal breakup occurs at the undeformed tip of the overriding plate, resulting in a highly asymmetric final rift-geometry.



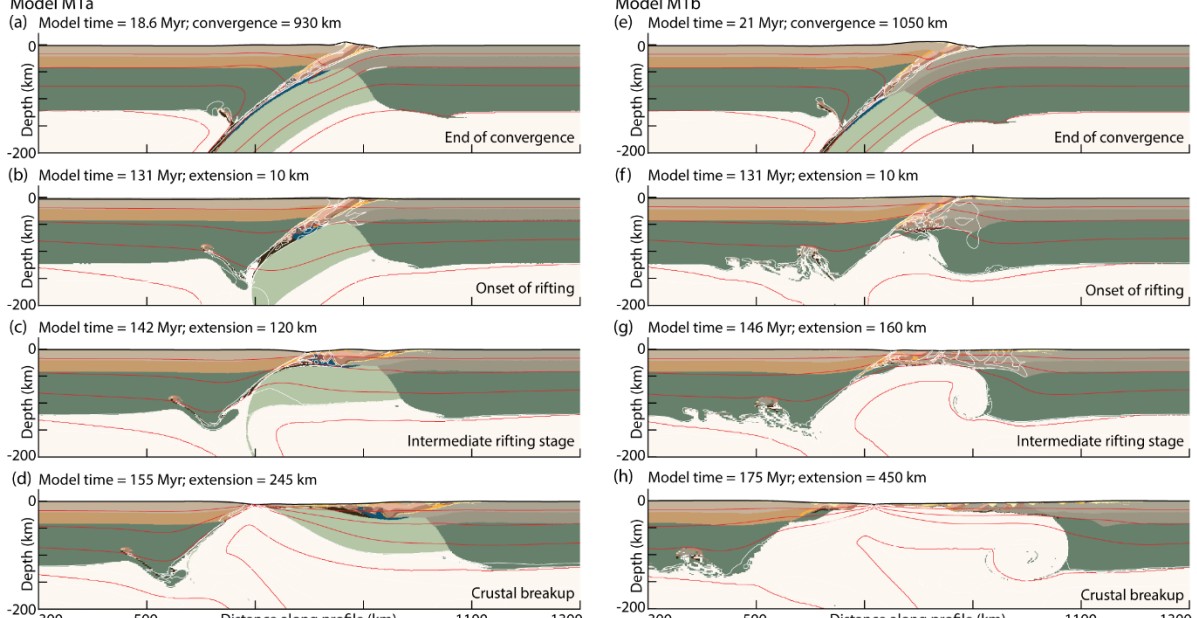

**Figure 5. Models M1a and M1b with one accreted terrane. (a-d) M1a characterized by a small orogen, exhibiting full inversional reactivation of the oldest subduction suture and only minor reactivation of the younger suture during rifting; (e-h) M1b characterized by a large orogen, exhibiting strong inversional reactivation of both subduction sutures and the formation of new normal faults, with final rifting away from the sutures in the hot orogen. Material colours (see legend of Fig. 1) at key time steps, with selected isotherms displayed by red contours (300°C, 600°C, 900°C, and 1300°C) and areas experiencing high strain rates displayed with white contours ($10^{-13}$ to $10^{-15}$ s$^{-1}$).**

The relaxation phase of experiment **M1b** (Fig. 5e-h and Supplementary Animation 5; for viscosity plots see Fig. C3e-h) is very similar to that of **M0b**, as the slab that hangs below the orogen is slowly wrenched off by its own weight with the detachment forming below the accreted terrane but within the incoming continental portion of the lithosphere. The result is a dome-shaped asthenospheric upwelling in the lithosphere, below the thickened (70 km) crustal root that consists of accreted terrane crust and incoming continental crust, supporting a 4.5 km-high, 200 km-wide, hot orogen where Moho temperatures locally exceed 900 °C. Once more, similarly to experiment **M0b**, all three types of inheritance are present at the onset of rifting. Structural inheritance includes two sutures and a major fault dissecting the accreted terrane, rheological inheritance is represented in the accreted terrane itself and the thickened crust of the adjacent incoming plate, and thermal inheritance manifests through a high geothermal gradient across the central orogen. Extension during the rifting phase initiates in the incoming continental portion of the orogen and the inherited sutures at the roof and sole of the accreted terrane are reactivated only after the orogenic crust is thinned out. In the last 30 Myr of the experiment extension is accommodated along several shear-zone pairs simultaneously, resulting in the formation of a wide margin on the incoming plate side, with graben-like structures, dipping towards the rift-axis. Full crustal breakup occurs close to the largely undeformed overriding plate resulting in a highly asymmetric final rift-geometry.



The relaxation phase of experiment **M2a** (Fig. 6a-d and Supplementary Animation 6; for viscosity plots see Fig. C4a-d) is very similar to those of **M0a** and **M1a** with the necking of the slab at a depth around 200 km and a long sliver of oceanic lithosphere preserved at the sole of the two accreted terranes, dipping towards the overriding plate. In this experiment, the necking process results in an almost 300 km-wide, 2 km-high orogen that lacks a proportional, thickened crustal root and is consequently slightly colder than its surroundings. As with experiments **M0a** and **M1a**, the main types of inheritance present at the onset of

extension are structural and rheological. In this case, three sutures, and two accreted terranes, underlain by oceanic mantle-lithosphere represent them each. The rifting phase evolves similarly to that of experiment **M1a** with extensive inversional reactivation of the oldest subduction interface focusing most of the extensional deformation on the highly deformed leading edges of the accreted terranes and creating an asymmetric rift with the narrow margin on the overriding plate side. Rifting occurs at the undeformed tip of the overriding plate, resulting in a highly asymmetric final rift-geometry.

Finally, the relaxation phase of experiment **M2b** (Fig. 6e-h and Supplementary Animation 7; for viscosity plots see Fig. C4e-h) is very similar to those of **M0b** and **M1b** as the slab that hangs below the orogen is slowly wrenched off by its own weight with the detachment forming below the accreted terranes but within the incoming continental portion of the lithosphere. Once again, the result is an asthenospheric upwelling in the lithosphere, under a thickened (80 km) crustal root that consists mostly of the juxtaposed accreted terranes and incoming continental crust, supporting a 4.5 km-high, 300 km-wide, hot orogen, where

Moho temperatures locally exceed 900 °C. Prior to the onset of rifting, all three inheritance types are present prominently. Structural inheritance includes three sutures, rheological inheritance is represented by the two accreted terrane and the thickened crust of the adjacent incoming plate, and thermal inheritance manifests through a high geothermal gradient across the central orogen. In this experiment, extension initially focuses on the suture separating the two microcontinents, but eventually heavily deforms the whole orogen, creating a wide margin with ribbon-like continental fragments sitting directly

on sublithospheric mantle on the overriding plate side. On the subducting plate side, the margin includes an approximately 150 km wide hyper-extended zone of sediments with relatively narrow proximal and necking domains.





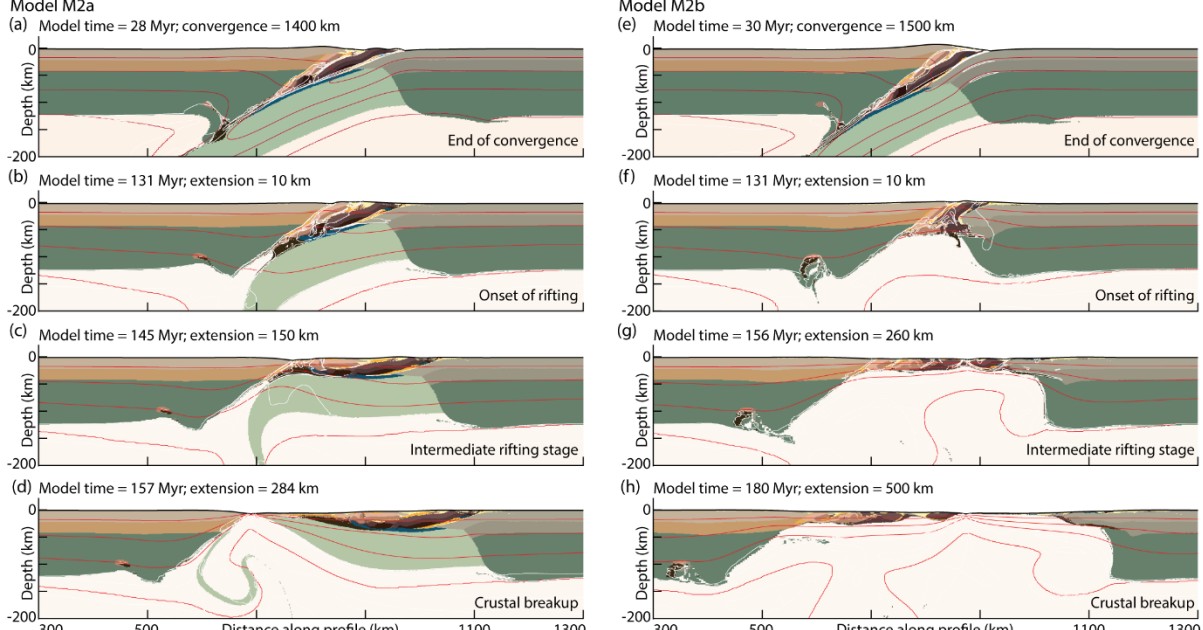

**Figure 6. Models M2a and M2b with two accreted terranes. (a-d) M2a is characterized by a small orogen, exhibiting full inversional reactivation of the oldest subduction suture during rifting and only partial reactivation of the younger ones; (e–f) M2b is characterized by a large orogen, exhibiting inversional reactivation of all subduction sutures during rifting accompanied by the formation of new normal faults and final rifting along the youngest suture, close to the centre of the hot orogen. Material colours (see legend of Fig. 1) at key time steps, with selected isotherms displayed by red contours (300°C, 600°C, 900°C, and 1300°C) and areas experiencing high strain rates displayed with white contours ($10^{-13}$ to $10^{-15}$ s$^{-1}$).**

## 4. Discussion

### 4.1. The relative importance of structural, rheological and thermal inheritance

Our numerical models of subduction, microcontinent accretion, continent-continent collision and subsequent rifting include the effects of three types of orogenic inheritance (defined after Manatschal et al., 2015) on the large-scale evolution of rifted margins (Fig. 1): (1) **rheological inheritance**; represented by variations in orogen size which result in differing orogenic root thicknesses and variations in the number of accreted terranes; (2) **structural inheritance**; primarily represented by the number of subduction zone sutures; and (3) **thermal inheritance**: where differences in orogen size lead to varying magnitudes of thermal anomalies. Here, we consider orogens to be small, when the incoming continental crust has not experienced significant under-thrusting by the end of the convergence phase. This corresponds to locally thickened orogenic crust in a zone no more




than 150 km in width with a corresponding 300 km wide topographic expression with a peak elevation of no more than 4 km at the onset of rifting (for the "largest" of the small orogens see model **M2a**, see Fig. 6b).

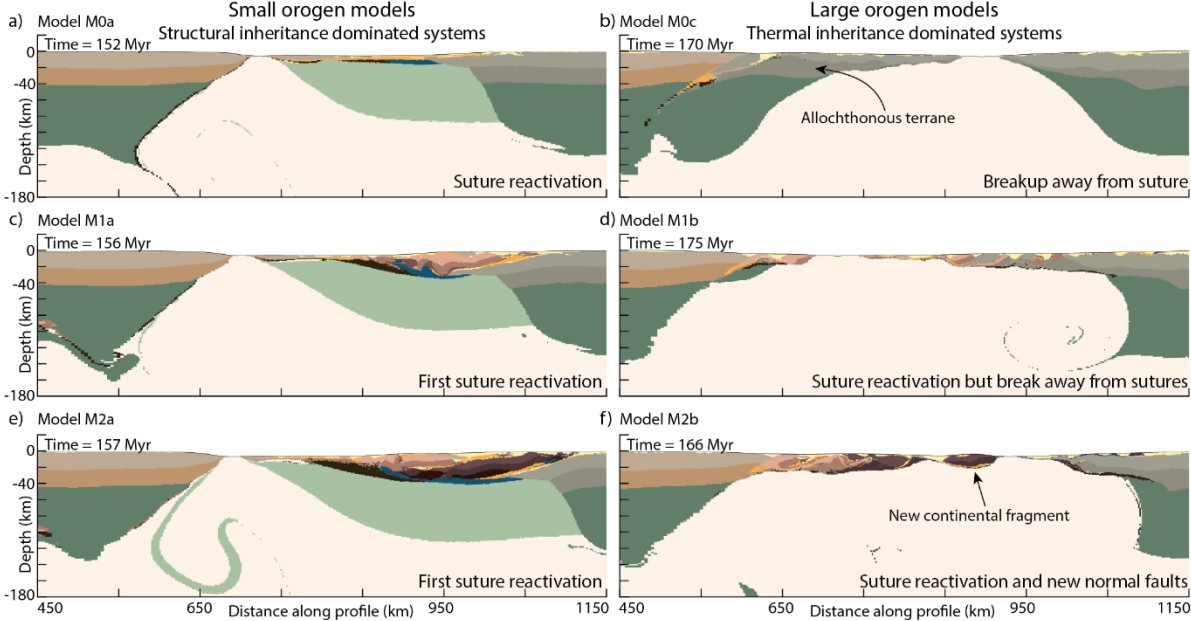

**Figure 7. Comparison of the final rifted margins of 6 experiments forming a small size orogen (left column, panels a, c and e) or large size orogen (right column, panels b, d and f) accreting 0 (top row, a and b), 1 (middle row, c and d) or 2 (bottom row, e and f) microcontinents. The comparison demonstrates the strong effect of the pre-rift thermal state of the lithosphere on the degree of suture reactivation and the final geometry of the resulting rifted margins. From a to f the snapshots are from models M0a, M0c, M1a, M1b, M2a and M2b.**

We find that the size of the orogen has a first order effect on the evolution of rifted margins (Fig. 7). During rifting of small orogens, deformation tends to exclusively localize on the inherited sutures, reactivating them as large-scale normal shear zones, while in large orogens deformation tends to localize away from the sutures in the orogen, creating new major extensional shear zones, rooted in the ductile lower crust (see each row in Fig. 7). This indicates that structural inheritance dominates over rheological and thermal inheritance in systems characterized by small, cold orogens, but that structural inheritance is of secondary importance in systems characterized by large, warm orogens. Increasing the degree of structural inheritance through the addition of more sutures (1 or 2 accreted terranes) does not have a dominant effect on the first-order rifted margin architectures even in the systems characterized by small, cold orogens (see the similar behaviour of the small orogen models in the first column in Fig. 7). Multiple sutures do, however, affect the localization on a smaller scale and the behaviour of individual structures as well as the preservation of microcontinent remnants in the rifted margin. When multiple sutures are present, the oldest, shortest and most favourably-oriented suture is reactivated most extensively, with the others experiencing only limited normal faulting. Reactivation of the sutures happens even in systems characterized by large, hot orogens. In





experiments **M1b** and **M2b**, where a large orogen contains multiple sutures, all sutures reactivate to some degree, with those that are closest to the centre of the orogen experiencing the most pronounced extensional inversion.

It is difficult to distinguish between the effects of rheological and thermal inheritance in our experiments, as the size of the orogenic root and the size of the positive thermal anomaly are directly related and both influence the system by lowering the strength of the lithosphere. The accreted terranes themselves represent an additional degree of rheological inheritance that is independent of the thermal evolution of the orogen prior the onset of rifting. The fact, that the large-scale deformation pattern is similar in models with and without accreted terranes when controlling for orogen-size (see the two columns in Fig. 7),

suggests that thermal inheritance plays a more prominent role here than rheological inheritance. However, on a more local scale there are notable differences. For example, exhumation of lower-crustal material to the surface only occurs in the presence of accreted terranes (see the proximal domain for models **M1a** and **M2a** and scattered in the distal domain for models **M1b** and **M2b**; Fig. 7). This exhumation seems to be facilitated by the inversion of the sutures for the small, cold orogen models, but there is no clear connection between the location of exhumed lower crust and sutures in case of the large, warm orogens.

These results must be interpreted cautiously, as there are other factors not accounted for, that might affect the relationship between rheological and thermal inheritance, such as the removal of crust that is rich in heat producing elements through efficient surface processes or the depletion of the crust after the end of orogeny. Both of these processes would result in a colder orogen at the onset of rifting and might enhance the role of rheological inheritance that is primarily of compositional origin.

The experiments presented above initially suggest that the kinematics of slab breakoff could play a pivotal role. Wrenching off of the slab after subduction results in a persistently shallow compositional LAB, although the thermal LAB mostly subsides to over 120 km depth by the onset of rifting (see models **M1b** and **M2b**; Fig. 4e-h and Fig. 5e-h). Despite the quick equilibration of the thermal LAB, the continental crust in these models heats significantly, due in part to the temporary proximity of hot asthenosphere and, more importantly, to the thickened orogenic crust which enhances radiogenic heat production. We propose

that this crustal thickening by continent-continent collision is the primary factor driving the high temperatures observed. For instance, in experiment **M0c** slab breakoff does not result in an asthenospheric upwelling, yet the orogenic crust behaves similarly to that in **M1b** and **M2b** (see the second column of Fig. 7 and Supplementary Animations 3, 5 and 7). It shows strong lower-crustal ductile flow that progressively widens the orogen, with central Moho temperatures exceeding 900 °C (Fig. 4j). An additional experiment with two microcontinents undergoing 34 Myr of convergence before thermal relaxation further

supports this view (see Fig. C1 and Supplementary Animation 8; for viscosity plots see Fig. C5). Here, slab breakoff mirrored that of **M0c**, and the resulting rift phase resembled **M2b** (compare Fig. 6h to Fig. C1d). This suggests that the thermal state at onset of rifting is primarily governed by orogen size rather than slab breakoff style, assuming similar thermal relaxation periods.

    In summary, structural inheritance is predominant when rifting takes place in small, cold orogens and thermal and rheological

inheritance dominate over structural inheritance in large, warm orogens. Assessing the relative importance of thermal and rheological inheritance is challenging, as they are linked through the size of the pre-rift orogen. However, the addition of



rheological inheritance in the form of accreted terranes has only a secondary effect, suggesting that thermal inheritance is the dominant factor of the two (Fig. 8).

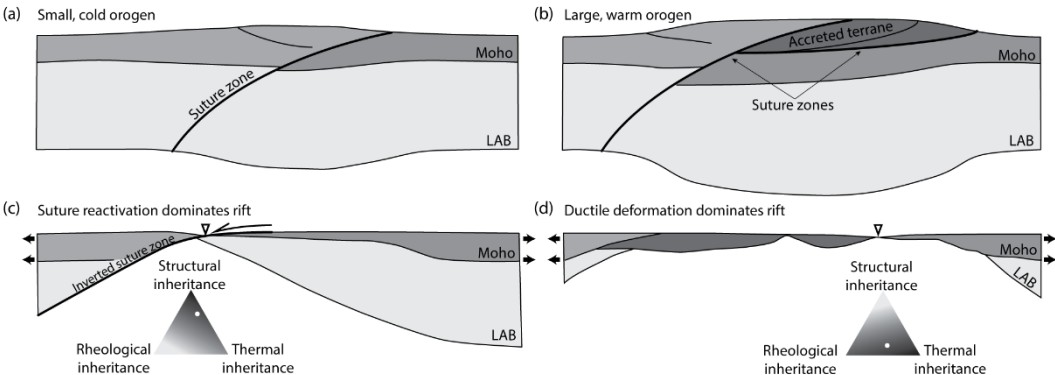

**Figure 8. Conceptual figure showing the difference in evolution between rifting a small, cold (a, c) and a large, warm (b, d) orogen. Ternary diagrams on (c) and (d) display the estimated relative importance in the rift evolution of the three inheritance types considered in this study.**

### 4.2.    Origin of thermal inheritance

Manatschal et al. (2015) considered two components of thermal inheritance from a previous phase of orogeny, that can play a significant role in determining its effects on rifting: 1) inherited heat and 2) potential of heat production. Inherited heat decreases with time and is consequently a function of the age of the lithosphere. This implies that the older the lithosphere, the colder the thermal structure and the stronger the lithosphere. In contrast, the potential of heat production depends on the presence of radiogenic elements, which is mainly controlled by the composition and age of the crustal rocks, the thickness of the heat-producing layer (that is also influenced by the rate of erosion and the time elapsed since the end of the orogeny), depletion processes and any magmatic additions. These factors taken all together imply that the degree of thermal inheritance depends on the time elapsed since orogeny in a very complex way.

A compilation of rifted margins by Buiter and Torsvik (2014) shows no correlations between suture age and rifting/breakup age, indicating that no simple relationship exists between thermal weakening and onset of rifting. However, as Buiter and Torsvik (2014) point it out, the lack of correlation does not imply that inherited faults would be the main localizing mechanism for extension at sutures. Local dominance of any of the three types of weakening mechanisms identified by Manatschal et al. (2015) in different places and times could result unconnected suture ages and rifting/breakup ages (Buiter and Torsvik, 2014). Our results show that collisional orogeny can be followed by a protracted phase of slab-detachment and orogenic collapse, accompanied by lower crustal flow, all of which act towards dynamically altering the thermal state of the lithosphere. Moreover, as seen in the experiments characterized by small orogens at the end of collision, an orogen might be colder than a thermally equilibrated lithospheric structure of it would dictate, as a subducting slab can carry cold lithospheric material to great depths (see depressed isotherms at the end of the orogenic phase of experiments **M0a**, **M1a** and **M2a**; Fig. 3b, 4b and 5b). In such a scenario, the lithosphere gradually heats over time and thermal inheritance becomes more and more prominent,




primarily within the lower crust. These numerical results all indicate that the thermal state of the lithosphere prior to rifting in the presence of orogenic inheritance depends very strongly on the particular dynamics of the orogenesis and should not be

viewed merely as a simple competition between two processes.

### 4.3. Comparison with previous modelling studies

Butler et al. (2015) used a similar setup to the one used here to model the evolution of the Mid-Norwegian rifted margin, characterized by narrower oceanic basins between microcontinents, a shorter period of quiescence between orogeny and rifting and a shorter phase of extension that did not result in full crustal breakup of the presented models. They varied the strength of

the incoming continent, essentially controlling for thermal and structural inheritance and focused on its effect on the exhumation history of the accreted terrane with a special goal of explaining the origin of ultra-high-pressure rocks found in the Western Gneiss Region. They found that the strength of the continental crust has an effect on the degree of internal deformation during orogeny but the extensional phase of their models was not long enough to produce rifted margins that we could compare our models with.

Petersen and Schiffer (2016) argue that the presence of a suture and a hydrated wedge above the shallow part of the subducted slab have a strong control on rift architecture. In their models, the suture seems to activate as a major detachment but as it is only defined within the mantle-lithosphere, it does not break the crust, and instead a set of new normal faults localize deformation. This setup preordains a strong asymmetry of the rifted margins which is also a persistent feature of the small orogen models presented here, where the presence of the tilted subduction interface engrains a degree of asymmetry. In our

models with large orogens, however, the suture plays a less prominent role and there is a freedom for new, opposite-dipping normal shear-zones to form, resulting in slightly more symmetrical rifted margins. The most symmetric rifting occurs in experiment **M0c** (Fig. 4i-j), where rifting largely happens away from the suture and conjugate shear-zones are dominating the rifting process. In general, based on the presented experiments, we argue that the more dominant the thermal inheritance is, the more symmetrical the rifting process becomes.

The orogens produced prior to the onset of rifting in the models of Salazar-Mora et al. (2018) are largely asymmetrical, generating a suture zone similar to those in our models with an outward propagating sequence of thrusts rooted in a mid-crustal detachment on the down-going side of the lithosphere. However, in their experiments there is no thermal relaxation phase that would allow for the heating of the lithosphere, resulting in a lack of thermal weakening and the absence of an associated lower-crustal ductile flow. As a result, reactivation of the thrust faults in an extensional mode during rifting is more prominent, even

in experiments where the orogen is large. The final rift location in their experiments tends to be close to the oldest, steepest and shortest upper crustal thrust fault; close to the centre of the orogen. The further away a thrust is from the centre of the orogen, the less propensity it has for extensional reactivation. These observations agree with the model results presented here, especially when looking at the small, cold orogens, where the role of structural inheritance outweighs that of thermal inheritance. In particular, we observe in model **M2a** (Fig. 6a-d) that the sutures further away from the centre of the orogen

experience significantly less inversional reactivation. Subsequently, Salazar-Mora and Sacek (2023) used a different numerical





model, but with a very similar setup, to explore the effects of tectonic quiescence between orogeny and rifting and found that the length of the quiescent period is important, when the orogen is large. In particular, their model M100 shows a phase of lower crustal ductile flow during the quiescence period and a subsequent necking of the mantle lithosphere away from the inherited suture. This result fits very well with our results from experiment **M0c** (Fig. 4i-j). This suggests that in cases where

no accreted terranes are involved, modelling only the continental collisional phase, without the prior phase of oceanic subduction, can be sufficient to capture the first order effects of both structural and thermal inheritance on rifting.

Chenin et al. (2019) explored the effects of wide-spread, thick mafic underplating at lower-crustal level below inherited crustal structures of an older orogen on a subsequent phase of rifting and found that such underplating in relatively cold lithosphere can suppress reactivation of crustal scale inherited structures. However, when the Moho temperature is high, this barrier cannot

function any longer. The experiments presented here do not produce similarly thick mafic underplating that could strengthen the lithosphere below shallow inherited structures, making a comparison difficult. What is clear, is that the mechanism proposed by Chenin et al. (2019) would act against the effects of the thermal weakening explored here, pointing towards a competition between rheological and thermal inheritance, where these features have competing effects.

In the models of Peron-Pinvidic et al. (2022) that included inheritance, the slab breaks off at the LAB and a largely continuous

LAB is formed by the end of the orogenic collapse phase. This corresponds well to our experiment **M0c** (Fig. 4i-h; and partly to the supplementary experiment **M2c**; Supplementary Fig. S1). In these models, final rifting occurs where the lithosphere was the thickest at onset of rifting. The multi-phase extension history applied by Peron-Pinvidic et al. (2022) has a notable influence on the exact rifted margin architecture, but the rift location remains unaffected. This supports our argument that the extension velocity has a secondary influence on overall rift architecture, compared to that of the state of the lithosphere prior the onset

of rifting.

### 4.4.    Rifted continental fragment formation

One of the most interesting aspects of our results is the formation of large allochthonous terranes and continental fragments during continental rifting. In experiment **M0c**, the rift forms in the original incoming plate, leaving a large portion of the orogen that has an incoming plate affinity, locked to the overriding plate in the footwall of the subduction suture (Fig. 7b). The

resulting allochthonous domain has an approximately 80 km-thick mantle-lithospheric root and a crustal thickness of approximately 40 km. In experiments **M1a** and **M2a**, the accreted terranes both remain attached to the incoming plate at the end of rifting, sitting on top of preserved oceanic mantle lithosphere, and exhibit a maximum crustal thickness of approximately 30 km that gradually thins towards the rifted margins (Fig. 7c and e). In experiment **M1b**, most of the accreted microcontinental terrane is fragmented and scattered along the rifted margins, but a relatively large block remains attached to the overriding

plate, providing the bulk of the rifted margin architecture. Finally, in experiment **M2b**, both accreted terranes remain attached to the overriding plate, as a heavily folded crustal assembly in the rifted margin. Here, in the distal zone, a part of the second microcontinent forms a distinct crustal fragment that is separated from the continental margin by a sedimentary basin above exhumed mantle (Fig. 7f).



The formation of allochthonous terranes is a staple of the continued assembly and disassembly of continents on Earth. When
rifting occurs in a former collisional orogen the suture is rarely reactivated precisely enough to avoid leaving imprints on rifted
margin structures. This imprecision, as seen during the formation of the North Atlantic Ocean, provided the basis for Tuzo
Wilson's concept of cyclic plate behaviour (Wilson, 1966). Our results show that during rifting of a large, warm collisional
orogen deformation is more likely to localize away from the suture than in a small, cold orogen. This result is also in line with
the experiments produced by Salazar-Mora and Sacek (2023).

It is generally agreed that microcontinents and continental fragments mostly form in divergent tectonic settings and that such
events are particularly frequent during breakup of supercontinents (Molnar et al., 2018; Tetreault and Buiter, 2014; Torsvik et
al., 2013; Whittaker et al., 2016). Microcontinent and continental fragment formation has mainly been associated with
processes that are either 3D in their nature, such as oblique rifting and rifting in the presence of large inherited structures
(Molnar et al., 2018; Nemčok et al., 2016) or connected to rising plumes and their interplay with rifted margins (Müller et al.,
2001). Our experiments add an alternative mechanism of continental fragment formation to the list, by showing that their
separation from the rifted margins during rifting is also likely in the presence of previously accreted terranes within the
lithosphere (see model **M2b**, Fig. 7f).

## 5.      Conclusions

Our numerical models provide insights into the interplay between rheological, structural, and thermal inheritance caused by
subduction, terrane accretion, and continental collision on the evolution of rifted margins (Fig. 8). Our findings suggest that
the size and thermal state of an orogen prior to rifting exert a primary control on margin architecture (Fig. 7). In systems with
small, cold orogens, structural inheritance—particularly the presence and orientation of sutures—plays a dominant role,
leading to focused extensional reactivation of subduction sutures and asymmetrical rifting patterns. When multiple sutures are
present, the oldest, steepest and shortest suture tends to accumulate most of the extension. In contrast, large, warm orogens
promote the formation of new shear zones away from inherited sutures, resulting in a more symmetrical rift architecture.
Nevertheless, the inherited sutures also get inverted, but to a lesser degree than in small, cold orogens. This indicates a
transition in the role of structural inheritance, which decreases in importance as thermal – and to a lesser extent rheological –
inheritance intensifies. This interplay also changes with time, as thermal inheritance is by its nature time-dependent, whereas
structural inheritance is much less so.

Thermal inheritance, largely governed by orogen size and the dynamics of slab detachment and lower crustal flow during
orogenesis, is critical in determining the thermal state of the lithosphere at the onset of rifting. Thick orogenic crust amplifies
radiogenic heat production, creating conditions favourable to thermal weakening and symmetrically distributed faulting. This
dynamic evolution of the thermal state, influenced by multiple processes, implies that the role of inheritance can only be
properly accounted for through assessing the thermal and mechanical history of the lithosphere prior to rifting.



Our models also highlight a potential new mechanism for allochthonous terrane and continental fragment formation. Previous studies have primarily attributed their formation to 3D rifting processes or interactions with mantle plumes, but our results show that the presence of accreted terranes within the lithosphere alone can facilitate their formation and detachment during rifting.

## 6. Appendices

**Appendix A – Numerical model description – SULEC v.4**

SULEC is a two-dimensional finite-element arbitrary Lagrangian-Eulerian numerical code that solves the plane-strain incompressible momentum equation for slow viscous-plastic creeping flows:

$$\nabla \cdot \sigma' - \nabla P + \rho \boldsymbol{g} = 0 \tag{1}$$

$$\nabla \cdot \boldsymbol{u} = 0 \tag{2}$$

where σ' is the deviatoric stress tensor, $P$ is dynamic pressure, $\rho$ is density, **g** is gravitational acceleration (9.81 m s$^{-2}$ in the vertical direction) and **u** is the velocity vector. Pressure is calculated using the iterative Uzawa formulation (Pelletier et al., 1989):

$$P^i = P^{i-1} - f_c \nabla \cdot \boldsymbol{u}^i \tag{3}$$

where $i$ signals pressure iteration number (with a cap of 75 iterations) and $f_c$ is the compressibility factor, which is 4 orders of magnitude greater than the maximum allowed viscosity in our models. The code relies on the Boussinesq approximation of small changes when describing the temperature dependence of density:

$$\rho = \rho_0 \big(1 - \alpha(T - T_0)\big) \tag{4}$$

where $\alpha$ is thermal expansion and $\rho_0$ is reference density at temperature $T = T_0$.

For viscous deformation a nonlinear, thermally activated power law creep formulation is used:

$$\sigma'_{eff} = f \cdot A^{-1/n} \cdot \dot{\varepsilon}'^{1/n}_{eff} \cdot w^{-r/n} \cdot d^{m/n} \cdot e^{\frac{Q+PV}{nRT}} \tag{5}$$

where $f$ is an optional scaling factor, $A$ the power law pre-exponent, $n$ the power law index, $\dot{\varepsilon}'_{eff}$ the effective strain rate, $w$ the water content, $r$ the water-content exponent, $d$ grain size, $m$ the grain-size exponent, $Q$ the activation energy, $V$ the activation volume, and $R$ the universal gas constant (8.314 J mol$^{-1}$ K$^{-1}$). Effective stress and strain rate are defined as:

$$\sigma'_{eff} = \left(\frac{1}{2} \cdot \sigma'_{ij} \cdot \sigma'_{ij}\right)^{\frac{1}{2}} \tag{6}$$

$$\dot{\varepsilon}'_{eff} = \left(\frac{1}{2} \cdot \dot{\varepsilon}'_{ij} \cdot \dot{\varepsilon}'_{ij}\right)^{\frac{1}{2}} \tag{7}$$

with summation over repeated indices implied.

Plastic failure is approximated with the Drucker-Prager yield-criterion:

$$\sigma'_{eff} = P \cdot \sin\varphi + C \cdot \cos\varphi \tag{8}$$





where $\varphi$ is the angle of internal friction and $C$ the cohesion. This approximation is a smoothed version of the Mohr-Coulomb

yield-criterion in plane-strain. Strain weakening (e.g., Pysklywec et al., 2002) is accounted for by linearly reducing $\varphi$ through

a predefined effective plastic strain ($\varepsilon'_{eff}$) interval of $0.5 < \varepsilon < 1.5$.

Since the equations for density and viscous material behavior (4 and 5) are temperature-dependent, we also solve the heat-transport equation:

$$c_p \rho \left( \frac{\partial T}{\partial t} + \boldsymbol{u} \cdot \nabla T \right) = \nabla \cdot k \nabla T + H + H_{sh} \qquad (9)$$

where $c_p$ is the specific heat, $t$ the time, $k$ the thermal conductivity, $H$ the heat production and $H_{sh}$ the shear heating. Shear

heating is calculated from the effective stress and effective strain rate using the following equation:

$$H_{sh} = f_{sh} \cdot 2 \cdot \sigma'_{eff} \cdot \dot{\varepsilon}'_{eff} \qquad (10)$$

where $f_{sh}$ is the shear heating efficiency factor, set to be 1 in these experiments.

We implement a simplified eclogite phase-transition in the models of this study: when the oceanic crust and the overlying

sedimentary material reach the appropriate temperature–pressure conditions they are instantly transformed into eclogite. Here

we use the eclogite stability field of Hacker (1996). In this simplified approach, only the density of the transforming material

is adjusted (increased) during the phase-transition, while the flow-law remains the same and latent heat is not accounted for.

The same eclogite phase-transition is applied to the lower crust of the microcontinental terranes. We base this choice on the

common argument that the lower crust of oceanic plateaus, submarine ridges, island arcs, and microcontinents is an ultramafic

cumulate layer (Behn and Kelemen, 2006; Mann and Taira, 2004; Schubert and Sandwell, 1989; Tetreault and Buiter, 2014).

To solve the above system of equations for stress equilibrium, material behaviour and thermal structure (equations 1-10)

SULEC uses the direct solver PARDISO v8.0 developed for sparse matrices (Alappat et al., 2020; Bollhöfer et al., 2019;

Bollhöfer et al., 2020).

SULEC v.4 employs an Arbitrary Lagrangian-Eulerian discretization method that allows for small vertical deformation of the

quadrilateral elements with remeshing per time step in order to achieve a true free surface (Fullsack, 1995). Independently of

the surface-process algorithm used, SULEC applies a stabilization term to density interfaces such as the free surface that

corrects numerical overshoots (Kaus et al., 2010).

**Appendix B – Description of model evolutions in collision phase**

Note: the experiments presented here are a continuation of selected collisional models presented in Erdős et al., 2024 (in

review), hence the following model evolution descriptions are shortened versions of the descriptions for models *R0*, *1Mw* and

*2Mw* presented there.



### Appendix B.1 – M0 model set

In our reference **model experiment set M0** an "empty" oceanic basin located between two continents is subducting under the overriding continent driven by a constant 5 cmyr$^{-1}$ boundary condition set at the far end of the incoming continent. During subduction a small sedimentary accretionary wedge is formed in the subduction zone but neither the overriding nor the incoming continent is deformed. Upon collision, first the incoming continent underthrusts the overriding continent continuously following the oceanic lithosphere. the most intense localized deformation occurs along the continuous subduction interface until about 17.6 Myr, when the under-thrusting continental crust reaches over 120 km deep. At this point, a new thrust-sheet forms in the footwall of the previous thrust, anchored in the ductile deformation zone at the base of the upper crust. The experiment concludes shortly after this at 18 Myr.

### Appendix B.2 – M1 model set

**Model experiment set M1** has the same basic setup as set M0, but it has a microcontinental terrane embedded within the oceanic lithosphere. After subduction initiation, the microcontinent reaches the trench at 7.4 Myr, with deformation focused on the subduction channel. Once at the trench, deformation localizes simultaneously at the top and the bottom of the microcontinent. By 11.8 Myr the original subduction zone is gradually abandoned for the detachment zone at the microcontinent's base, resulting in a new trench at the oceanward margin of the accreted terrane. This new subduction channel, located within the lower crust, remains largely continuous, leaving the middle and upper crust intact. At 15.8 Myr, the incoming continental margin reaches the new trench, initiating continent-continent collision. The original subduction zone atop the accreted terrane is reactivated and deforms alongside the new subduction zone. As the incoming continent under-thrusts, the accreted terrane begins slow internal fragmentation. By the end of the experiment at 21 Myr, most of the terrane is buried under the overriding continental crust but remains largely intact at shallow depths.

### Appendix B.3 – M2 model set

**Model experiment set M2** has the same setup as the previous two sets but has two microcontinental terranes embedded within its oceanic lithosphere. The model evolution is virtually identical to the previous experiment set up to 15 Myr, when the accretion of the first terrane is concluded. At 16.8 Myr, the second microcontinent arrives at the trench and begins under-thrusting the accreted terrane, similar to the first microcontinent's earlier behaviour. During this accretion event the original subduction interface, the subduction interface between the two microcontinental terranes and a shear zone within the lower crust of the second microcontinent are all active, but the current subduction interface is dominant. Multiple deformation zones remain active until 22.6 Myr, when the new subduction interface at the bottom of the second terrane becomes the sole zone of deformation. At 24.8 Myr, the incoming continental margin reaches the trench, starting continent-continent collision. During this phase, the incoming continent under-thrusts the accreted terranes while previous deformation zones reactivate. By 28 Myr,





the first terrane is mostly buried beneath the original overriding plate, with only a small outcrop visible, while the trailing edge of the second microcontinent remains exposed over a 25 km transect. By the end of the experiment at 30 Myr, both accreted terranes are buried but remain largely intact at shallow depths.

**Appendix C – Supplementary model S1**

Experiment **SM1** belongs to the M2 model set, with two microcontinents but the Collision Phase is 34 Myr long, and results in an extra-large orogen (Fig. C8). At the end of the Relaxation Phase, the orogen displays a geometry similar to that of experiment **M0c**, with a largely continuous LAB, with a 400 km wide zone of thickened lithosphere and crust and a 275 km wide orogen. The style of rifting is very similar to that of experiments **M0c**, **M1b**, and **M2b**, with deformation localizing

within the orogen on multiple shear-zones and final rifting occurring away from the inherited sutures. Similarly to experiment **M0c**, final rifting takes place within the crust of the incoming continent. Traces of the accreted terranes are present in both rifted margins with the majority located in the overriding plate-side.





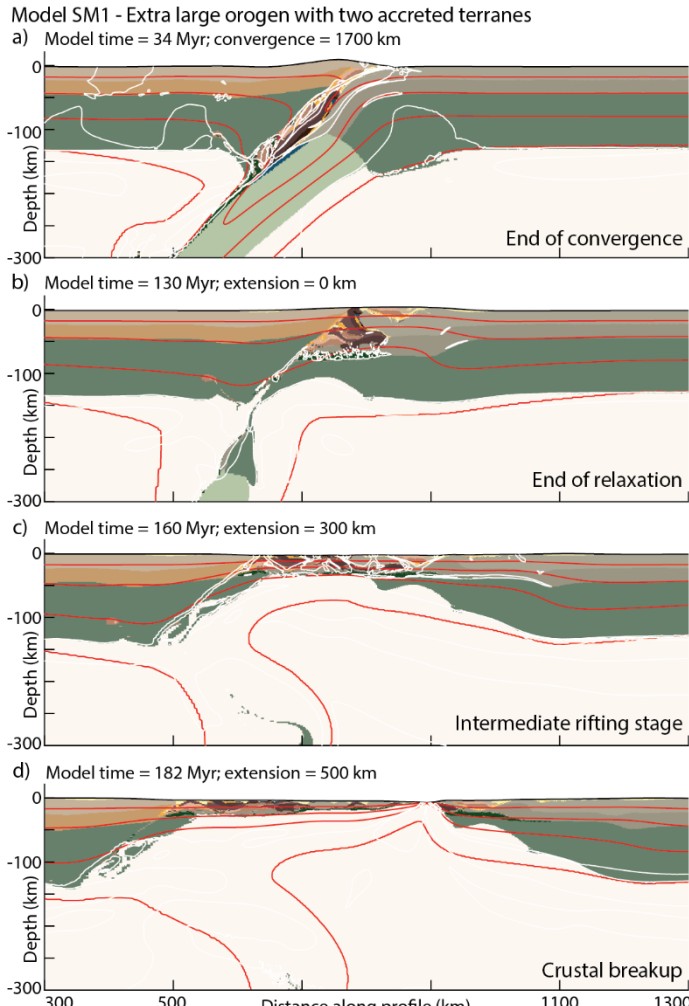

**Figure C 1. Model evolution of supplementary model S1. (a-d) SM1 is characterized by an extra-large orogen and slab breakoff below the LAB. During rifting, new normal shear-zones form within the orogen creating wide, asymmetric rifted margins. The style is similar to what is observed in experiments M0c, M1b and M2b.**



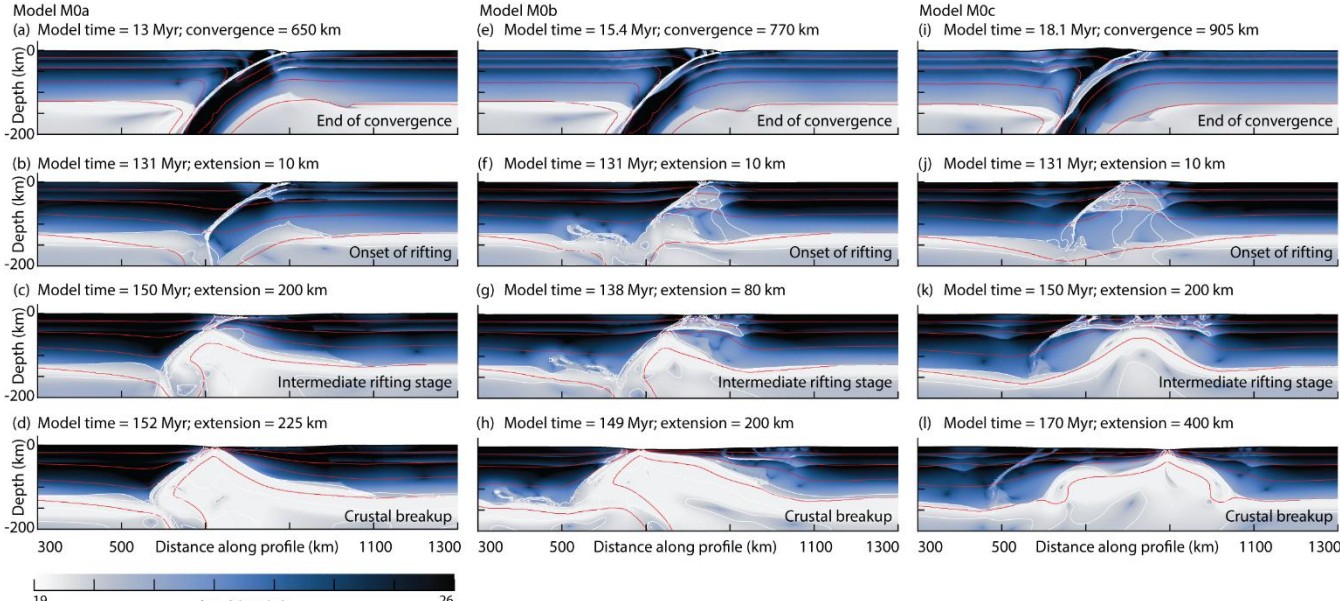

**Figure C 2. Viscosity snapshots of the evolution of model experiments M0a (a-d), M0b (e-h), and M0c (i-l). Selected isotherms displayed by red contours (300°C, 600°C, 900°C, and 1300°C) and areas experiencing high strain rates displayed with white contours (10⁻¹³ to 10⁻¹⁵ s⁻¹).**





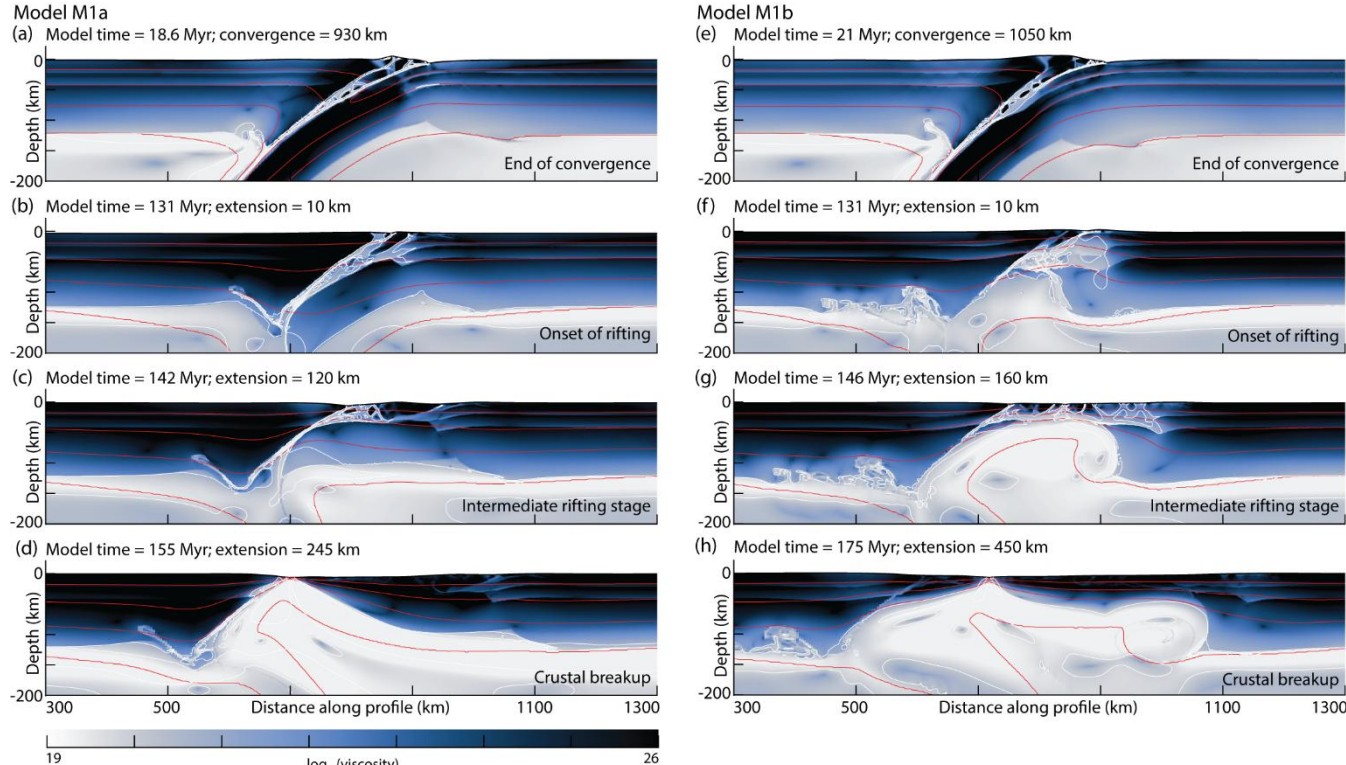

**Figure C 3. Viscosity snapshots of the evolution of model experiments M1a (a-d) and M1b (e-h). Selected isotherms displayed by red contours (300°C, 600°C, 900°C, and 1300°C) and areas experiencing high strain rates displayed with white contours ($10^{-13}$ to $10^{-15}$ s$^{-1}$).**



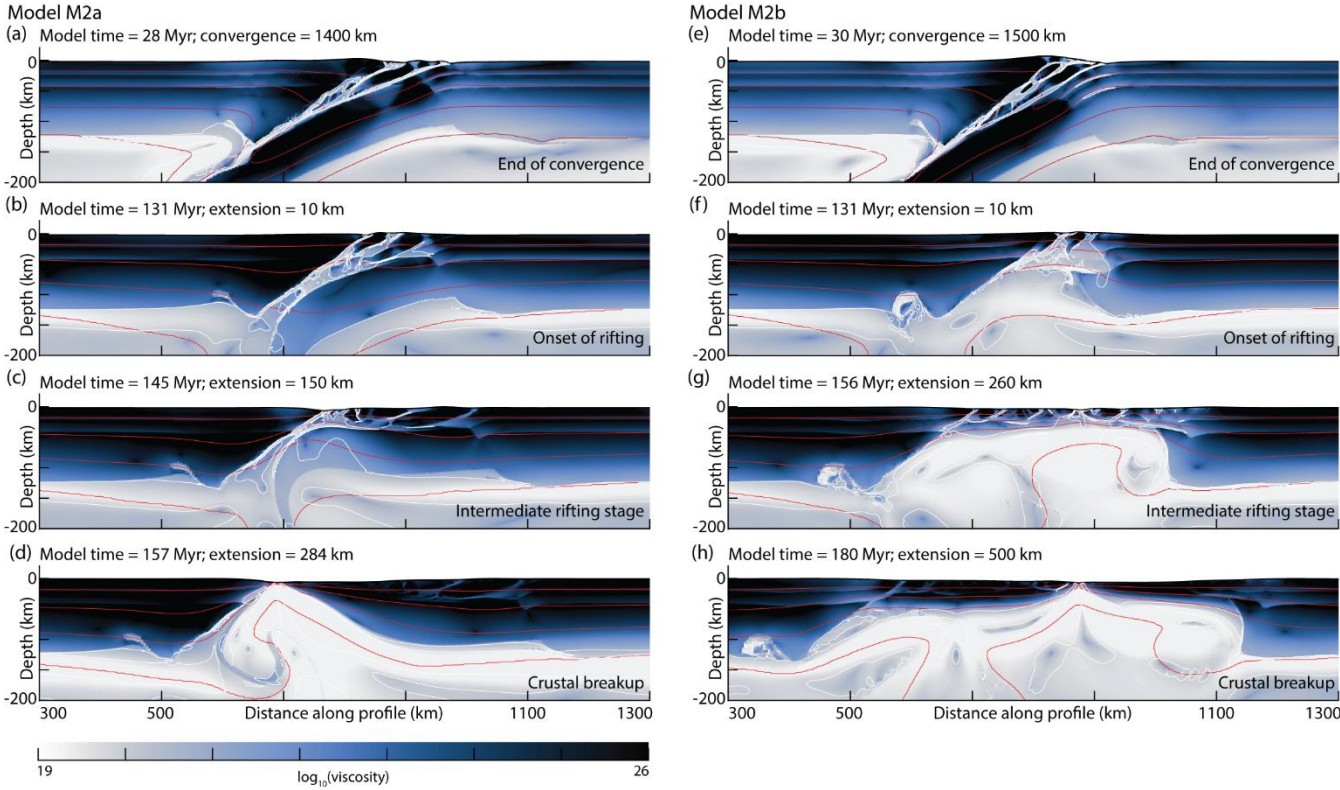

**Figure C 4. Viscosity snapshots of the evolution of model experiments M2a (a-d) and M2b (e-h). Selected isotherms displayed by red contours (300°C, 600°C, 900°C, and 1300°C) and areas experiencing high strain rates displayed with white contours (10⁻¹³ to 10⁻¹⁵ s⁻¹).**





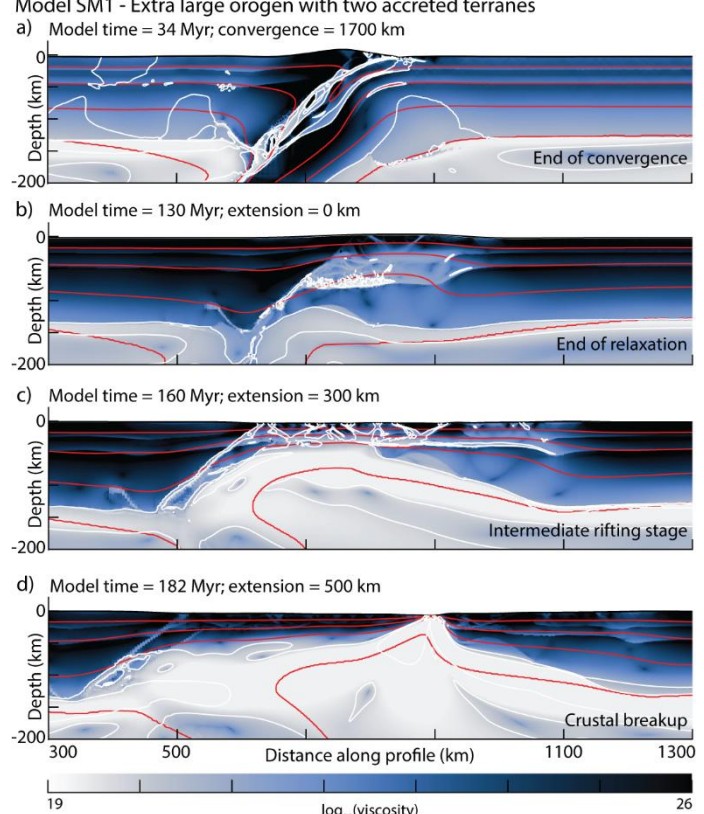

**Figure C 5. Viscosity snapshots of the evolution of supplementary model experiment SM1 (a-d). Selected isotherms displayed by red contours (300°C, 600°C, 900°C, and 1300°C) and areas experiencing high strain rates displayed with white contours ($10^{-13}$ to $10^{-15}$ s$^{-1}$).**

## 7. Code and Data availability

All data are available in the main text or the Supplementary Materials. Numerical models are computed with published

methods, described in the Methods section and Appendix A1. Animations of showing the evolution of the model experiments

along with the data used to generate the figures and animations is available on Zenodo (Erdős et al., 2025).

Current link for the depository:



## 8.      Acknowledgements

This study uses SULEC version 4, developed by S.B and Susan Ellis. A significant part of the calculations for this research were conducted with computing resources provided by GFZ. ZE would like to thank Ameha Muluneh for the stimulating
discussions on the East African Rift System.

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
