# Peer review of "The influence of accretionary orogenesis on subsequent rift dynamics"

_EGUsphere, 2025_

## Author Response (AR1)

**Reviewer 1:**

In their contribution, Erdős et al. use thermo-mechanical numerical modelling to investigate the impact of micro-continents incorporated in collisional orogens on subsequent rifting. They conclude that the structural inheritance related to the oceanic sutures are likely to control the location of future rifting and breakup when the orogen is small and cold because resulting from limited convergence. Conversely, when the orogen is large and hot consequently to protracted convergence, rifting and breakup are more likely to occur away from the suture zones.

The article presents new conclusions, which are sensible and consistent with observations and conceptual models. I find the article clear, well written and well referenced, including recent studies. The model parameters are clearly presented, and the model limitations are comprehensively listed. The results of the numerical experiments are discussed in the light of other numerical modelling studies. In my opinion, the paper can be accepted as is.

We thank the reviewer for their positive appraisal of our manuscript.

I have only two minor comments to the authors:

L 14: "the late-Paleozoic to Mesozoic opening of the Atlantic Ocean occurred immediately after the early Paleozoic Caledonian orogeny". In your numerical experiments, you consider a phase of thermal relaxation between the end of convergence and the onset of rifting. The Caledonian domain experienced presumably such a phase of relaxation between the end of the orogeny and the onset of rifting, often called "post-orogenic collapse" in the literature. Thus, I find your statement in L 14 misleading.

The Caledonian example is more complex than our experiments. The Silurian to Devonian Caledonian orogeny ended at around 400-390 Ma, followed by the very rapid exhumation of the Western Gneiss Region. This exhumation has been attributed to the slab detaching from the base of the orogen (Wiest et al., 2020 gives a nice review of timing of exhumation across the Western Gneiss Region). This again was followed immediately by a phase of orogenic collapse with large scale detachments still in the Devonian (Andersen and Jamtveit, 1990, Osmundsen et al., 2000, etc). Tectonic activity appears to have diminished during the Carboniferous. The reviewer is correct in pointing out that rifting and formation of the rifted margins does not start until the Permian-Triassic with the main phase in the Jurassic (Patruno and Archer, 2020 have a nice overview of the North Sea, which has similar timing to the mid Norwegian margin). There are inferences of minor normal faulting between the Devonian and the Permian, but there is a great deal of uncertainty in these deep offshore structures. We have taken the relaxation phase to be time passed between the end of orogeny and the onset of orogenic collapse. To avoid confusion, we have modified the sentence which now reads:

*"An excellent example of such a system can be found in the North Atlantic, where the late-Paleozoic to Mesozoic opening of the Atlantic Ocean followed the early Paleozoic Caledonian orogeny, that formed during the collision of Baltica and Laurentia continents but also incorporated allochthonous continental terranes."*

Figure 2: The figure displays cratons, Cenozoic rift basins and older rift basins while the figure caption announces that it displays "rheological, structural and thermal inheritance". At least explicit the correspondence between the two.

We thank the reviewer for pointing that out! We have made a more concrete connection between the inheritance types and the features presented on figure 2.

**Reviewer 2:**

The authors have conducted 2D thermo-mechanical experiments studying the effect of different types of inheritance on the evolution and dynamics of rifting. They have found that rheological and structural features play a major role in the rifting style of cold, small orogens, while thermal inheritance is most pronounced in warm, large orogens.

The study is novel and very informative, and sheds light on the very important (and often puzzling) aspect of inheritance. Moreover, the manuscript is very well presented. However, because it is essentially a continuation of the authors' submitted and not-yet peer-reviewed article (as of April 2025), it is not possible to assess this work independently. I recommend holding off on the acceptance of this paper until after the authors' submitted paper is accepted. Finally, to make this article work as a standalone study, without having to have read the submitted 2024 paper, I recommend appending the methodology with more details about some numerical decisions, values, etc., as well as more detailed description of the first phase, since most inherited features come from that first phase. I am presenting below a list of minor comments, which can hopefully speed up the review process should this article is re-submitted.

*As indicated in our initial response (of 28 April 2025), the manuscript upon which this work builds was preprinted in ESS Open Archive and has now been peer-reviewed and published open-access (10 April 2025) as Erdős, Z., Buiter, S. J. H., & Tetreault, J. (2025). The role of microcontinent strength and basal detachment in accretionary orogenesis: Insights from numerical models. Journal of Geophysical Research: Solid Earth, 130, e2024JB029509.* https://doi.org/10.1029/2024JB029509

*The review process of the paper did not require changes to the models and resulted in no changes to results this manuscript builds upon.*

*We have expanded both Appendix A to include more details on numerical modelling choices, we have expanded on the description of the shortening phases in Appendix B and we have updated the citations to match the published version of the paper detailing the shortening phase.*

Minor comments:

Figure 2: Neither of the three types of inheritance is mentioned in the caption/figure label.

*We thank the reviewer for pointing that out! We have made a more concrete connection between the inheritance types and the features presented on figure 2.*

L 72: Can you briefly describe what an "empty" ocean basin is?

*We have clarified this statement and now write:*

*"The original Wilson Cycle formulation focused on the recognition of earlier oceans preceding continental collision and subsequent ocean formation and was thus, for simplicity, described as an "empty" ocean basin, devoid of any and all anomalous structures."*

L 59-82: I understand that separating the rheological and structural inheritance is difficult, but the way I am reading the text, there is some rheological characteristics in the structural inheritance, especially when they are describing accretion and collision (thinking about the inherently weak sediments that can make up a subduction interface and how they are rheologically weaker and can thus localize deformation). Perhaps you could mention this in the paragraph about structural inheritance?

*This is a very good point. There is an element of rheological inheritance to the way the subduction sutures act as they incorporate weak material brought into the shear-zone. We have added this point in the introduction.*

L 87-89: Are there any studies regarding the geothermal gradient in that region that can answer this question?

The difficulty here is two-fold: (1) we would need to understand the thermal state of the lithosphere prior the onset of rifting, and (2) the presence of the hotspot tends to overprint the potential steady state geotherm of the lithosphere. We are unaware of observational studies addressing this complex problem in East Africa.

L 117: Why was there no localization at former suture zones?

This refers to the following text: *"Chenin et al. (2019) explored the effects of wide-spread, thick mafic underplating at lower-crustal level below inherited crustal structures of an older orogen on a subsequent phase of rifting. This setup incorporates both structural and rheological inheritance to simulates observations of the setting below the Variscan orogenic structures of Western Europe, where rifting during the opening of the Tethyan and the North Atlantic did not localize at former suture zones. Chenin et al. (2019) found that such underplating located in thermally equilibrated lithosphere can suppress reactivation of crustal scale inherited structures."*

To the best of our knowledge, this is an open question. The hypothesis put forward by Chenin et al. (2019) is that magmatic underplating within a thermally equilibrated lithosphere creates a strong zone that can shield the lithosphere from stresses by increasing its integrated strength. However, in their underplating model they have also suppressed the suture at lower crust under upper mantle depth by the mafic underplating body. We have clarified this in the introduction.

Table 1: Many of the parameters used are not explicitly mentioned in the text (I assume because they are part of the 2024 article under revision). For instance, why did you choose such low φ values for some layers, what do the small differences in the densities and cohesions of the crustal rocks reflect, where is eta_eff used?

We would like to thank the reviewer for pointing out that these choices need clarification. In particular, eta_eff is the effective strain that is used in tracking the amount of deformation experienced by the rocks and allows for – for example – strain weakening to be calculated and tracked. This information is already present in Appendix A. The small differences in density reflect well documented differences in average composition. The low cohesion and internal angle of friction values of the microcontinent gabbroic lower crust represent pre-existing weak zones carried over from the time of their formation. In the - now peer-reviewed and published - paper on the collision phase, we have demonstrated that such weak lower crust is needed in order for multiple terranes to accrete. To mimic these pre-existing weaknesses, the yield strength of this layer is lowered by the assignment of reduced values for cohesion and internal angle of friction. We have added this information in the Experimental setup section.

L 154: What is the significance of the scaling factor? I understand how it works mathematically, but why is this a necessary addition, what purpose does it serve?

We use the scaling factor (f) to approximate the effect of variations in volatile content and potential changes in strength due to minor compositional variations (e.g., Beaumont et al. 2006). In particular, laboratory data show that water-saturated olivine is 5–20 times weaker than dehydrated olivine at the same strain rate (Hirth & Kohlstedt, 1996; Karato & Wu, 1993). The wet olivine rheology of the continental mantle-lithosphere is scaled up (f=5) in order to supress instability at the bottom of the mantle-lithosphere that otherwise occurred during the long thermal relaxation phase. We have now clarified this in the text.

L 157: Using gabbro rheology for sediments is a bit unusual, so it would help if you could write an explanation for that.

That is actually a mistake in the text, and – as is correctly stated in Table 1 – the sediment has a rheology of wet quartz. We thank the reviewer for spotting it; we have now corrected the text.

L 162: Why did you choose this angle (and orientation) for the trailing continental margin? If we consider that the trailing continent was the result of rifting, one expects the margin to have the opposite orientation I think, but maybe I am missing something here.

The trailing continental margin architecture was designed to replicate the geometry of a very simple, narrow rifted margin. We decided against implementing a margin with a more complex internal structure in order to be able to focus on the effects of the inheritance cause by the collision rather than any arbitrary structural inheritance coming from a hypothetical previous deformation phase. Based on the reviewer's comment, we expanded on this modelling choice in the text.

L 176: I was wondering what the significance of the microcontinent widths is. Would your results be significantly different? And why this width?

We have not systematically tested the effects of varying microcontinent size, but that would certainly merit its own study. We have used the same size as Tetreault and Buiter (2012), whose models we were building on. Including size as an extra variable would make it difficult to separate the influence of multiple microcontinents accreting from size on structural inheritance. Nevertheless, we would argue that 200 km is a fairly representative size for microcontinents (see for example the Rockall plateau). Hypothetically, we would expect significantly wider microcontinents to produce more internal structures during accretion and provide more structural inheritance for the extensional phase. The width of the incoming microcontinent would likely change the overall buoyancy and internal strength and thus impact its fate in the subduction zone (that is, influence whether it collides, accretes, or subducts). Of course, size (and therefore buoyancy) is not the only factor influencing the fate of microcontinents and Tao et al., 2020 did a nice geodynamic numerical study comparing seamounts to oceanic plateaus in subduction zones and they showed that for some overriding accretionary wedge geometries or in case the lower crust of the FAT is "weak" you can still accrete large dense FATs.

L 204: How do you identify the time when deformation moves into the hyper-extension regime?

We first run the models at a constant, 1 cm yr$^{-1}$ extension velocity until break-up is achieved. Then we identify the time, when the extensional deformation localizes exclusively in the upper plate crust and begins a sequential migration of oceanward-younging, upper crustal normal faulting. This is a largely self-sustaining behaviour described by Brune et al. (2016). We then use the timestep before the onset of this process as a restart point and increase the extension velocity to 5 cm yr$^{-1}$ from this timestep onwards. We now include this description in the text as a response to this comment.

L 205-206: I assume these are typos and you meant M0a, M0b etc.?

Yes, they are. We thank the reviewer for picking this up.

L 208: Why specifically 130 Myr?

The length of the relaxation phase was chosen to allow for the slab to break off and the temperature field in the orogenic lithosphere to reach close to a steady state in all models. Here, by steady state, we mean that the depth of the 600 °C isotherm does not change by more than 100 m in 1 Myr. Based on the analysis of our models, the shallower isotherms are generally more stable in time while the deeper ones can move somewhat faster, but generally within the same order of magnitude. We thank the reviewer for pointing out that this information was missing from the text; we have now included it.

Paragraph 2.1: Just so one can have a sense of resources, how computationally expensive were the models?

The experiments were mostly run on the GFZ Linux cluster GLIC. The code is largely sequential, but the matrix solver is parallel and the compiled code scales well when the calculations are done on a single node. We used nodes with up to 12 cores on GLIC. The shortest run was M0a, and it used approximately 11000 CPU hours, while the longest run was M2b, that used approximately 25000 CPU hours. This boils down to about 40 days for the quickest model and up to 90 days for the slowest one.

L 231: Strain weakening is not mentioned in the numerical setup nor in the table. Perhaps add a short phrase?

The strain-weakened strength profiles are displayed in Figure 3b, while the method used is described on lines 575-576 in Appendix A. The strain-weakened values of the internal angle of friction are also displayed in Table 1, but they were not mentioned in the caption. Based on the reviewer's comment we amended the caption of Table 1.

Paragraph 3.1: It's great that you give a brief description of the models; I recommend summarizing them also in a table, e.g., Model name – inheritance – suture behaviour – etc.

We believe that the adding the keywords in a table would only duplicate the information that is already distilled in a concise manner in this section, making it repetitive. Keeping these descriptions in a text format allows us to present a bit more context.

L 256-259: Why is this additional material important?

This refers to the following text: *"This time was chosen because the additional convergence allowed for an amount of material equivalent to one microcontinent (of the size used in this study) to enter the subduction zone (i.e., large orogen). Finally, in experiment M0c, the shortening phase lasts until 18 Myr allowing for an amount of material equivalent to two microcontinents to enter the subduction zone (i.e., extra-large orogen)."*

This choice was made explicitly to ensure that the size of the orogens remains comparable across the model sets. It also helps maintain a consistent degree of thermal inheritance between them. In response to this comment, we have now clarified this point in Section 3.1.

L 269: Maybe add a short sentence introducing/describing what a large orogen is?

We thank the reviewer for the suggestion. We included a short sentence to expand on what we term small and large orogens.

Figure 7: For consistency, you could keep the subfigure names going vertically, instead of horizontally (same for Fig. 8).

Flipping the columns and rows in figures 7 and 8 would indeed provide consistency between the results and the discussion figures. However, for the changes to work, we would need to create a three-panel wide figure 7 and reduce the size of the cross-sections, which would reduce quality as well as comparability. There are no such concerns regarding figure 9, but we believe that keeping the discussion figures in a consistent format is also helpful for the readers. For these reasons we would prefer to keep the layouts as they are.

L 391-392: In the absence of healing, pre-existing structures also lower the strength of the lithosphere. Perhaps you could comment on this?

This is a valid point, but its discussion might be better placed with the discussion of the effects of multiple sutures on lines 380-385. We have expanded on the discussion there based on the reviewer's comment. The section now reads:

*"Increasing the degree of structural inheritance through the addition of more sutures (1 or 2 accreted terranes) does not have a dominant effect on the first-order rifted margin architectures even in the systems characterized by small, cold orogens (see the similar behaviour of the small orogen models in the first column in Fig. 7). This is true, even though strain healing is not accounted for in these models. This means that the inherited sutures, that have weakened due to strain accumulation through extensive deformation (by the gradual lowering of the internal angle of friction) remain weak, regardless of how long they might remain undeformed after the conclusion of the shortening phase. Multiple sutures do, however, affect the localization on a smaller scale and the behaviour of individual structures as well as the preservation of microcontinent remnants in the rifted margin."*

L 409-419: This might be a long shot, but are there any geological constraints on this, i.e. from the rock record?

This refers to the following text: *"We propose that this crustal thickening by continent-continent collision is the primary factor driving the high temperatures observed. For instance, in experiment M0c slab breakoff does not result in an asthenospheric upwelling, yet the orogenic crust behaves similarly to that in M1b and M2b (see the second column of Fig. 7 and Supplementary Animations 3, 5 and 7). It shows strong lower-crustal ductile flow that progressively widens the orogen, with central Moho temperatures exceeding 900 °C (Fig. 4j). An additional experiment with two microcontinents undergoing 34 Myr of convergence before thermal relaxation further supports this view (see Fig. C1 and Supplementary Animation 8; for viscosity plots see Fig. C5). Here, slab breakoff mirrored that of M0c, and the resulting rift phase resembled M2b (compare Fig. 6h to Fig. C1d). This suggests that the thermal state at onset of rifting is primarily governed by orogen size rather than slab breakoff style, assuming similar thermal relaxation periods."*

It is a long shot indeed, but an interesting one! There have been studies into the effects of slab breakoff during the closure of an oceanic basin and its impacts on the orogen. De Boorder et al. (1998) for example explained mineralization in orogenic belts at the end of orogeny by slab breakoff and consequent emplacement of hot asthenosphere at shallow levels which increases the heat flux into the continental lithosphere, causing generation of melts. There have also been suggestions that slab tear propagation can be tracked in the surface evolution of young orogens, such as in case of the Apennines (e.g., van der Meulen et al., 1998; Ascione et al., 2012). In the Caledonides, exhumation of the Western Gneiss Region due to slab detachment occurs largely as migmatitic core complexes (Wiest et al., 2020), which would indicate elevated temperatures and lower crustal flow. But whether any of these data could be used to differentiate between the types of slab detachment observed in our models is firmly beyond the scope of this study.

Figure 8: Great way to show the three inheritance types (ternary)! On a slightly separate note, I was wondering why rifting does not usually initiate elsewhere than the inherited structures, given the strain weakening mechanism in the models.

The strain weakening mechanisms only really affect the orogens, so localization will happen in the orogen. In the small, cold orogen models, rifting does initiate on the inherited sutures but in the large, warm orogen models it initiates away from the sutures, within the thickened crust.

L 443: What exactly is lower crustal flow? I might have missed it. Also, how/why does it alter the thermal state?

Lower crustal flow is the extensive ductile deformation of the warm, weak lower crust. In our experiments, it widens the orogenic root of the large, warm orogen models during the thermal relaxation phase. It also appears in the hyper-extension phase of the models when the extension

velocity is not increased (as in Brune et al., 2016). Based on the reviewer's question, we include a short explanation on line 210 in the track changes file.

L 472: Is this thermal relaxation phase defined somehow also rheologically?

This refers to the following text: *"The orogens produced prior to the onset of rifting in the models of Salazar-Mora et al. (2018) are largely asymmetrical, generating a suture zone similar to those in our models with an outward propagating sequence of thrusts rooted in a mid-crustal detachment on the down-going side of the lithosphere. However, in their experiments there is no thermal relaxation phase that would allow for the heating of the lithosphere, resulting in a lack of thermal weakening and the absence of an associated lower-crustal ductile flow"*

If the line-number indicated by the reviewer is correct, then this question pertains to the work of Salazar-Mora et al. (2018) that we have compared our results with. Since this is not our work we can only speculate, but the absence of a phase of thermal relaxation from their experiment also implies there is no effect on rheology. In our experiments, the relaxation phase is defined thermally. The effects on the rheology of the orogen vary with the thermal response of the lithosphere.

L 498-500: How certain are you about this? I know it is beyond the scope of this paper, but did you run any models with varying extension velocities? Any other relevant studies?

This refers to the following text: *"This supports our argument that the extension velocity has a secondary influence on overall rift architecture, compared to that of the state of the lithosphere prior the onset of rifting."*

Roger Buck's pioneering work has shown that for simple continental rifting in the absence of extensive inheritance, higher extension velocities should push the models towards more narrow-rifting mode. Since then, a large number of modelling studies focused on this topic and there is a general agreement on this result within the community. We have run these models with a very high (5 cm yr$^{-1}$) extension velocity as well and the first order features of the resulting rifts were largely identical, meaning that the large orogen models would still result in wide rifting. We have also run the large, warm orogen model with an extension velocity of 0.5 cm yr$^{-1}$ and the results were very similar in style to the experiment presented here, although the location of the final break-up was different. Nevertheless, we have not run a systematic analysis of the parameter-space as it is indeed beyond the scope of this study.

Paragraph 4.4: It would help if you added some labels on Fig. 8 that denote the accreted terranes.

We have added labels to figure 8 and indicated the terranes in the figure caption.

**References:**

Andersen, T. B., and B. Jamtveit (1990), Uplift of deep crust during orogenic extensional collapse: A model based on field studies in the Sogn-Sunnfjord Region of western Norway, Tectonics, 9(5), 1097–1111, doi:10.1029/TC009i005p01097.

Ascione, A., S. Ciarcia, V. Di Donato, S. Mazzoli, and S. Vitale (2012), The Pliocene-Quaternary wedge-top basins of southern Italy: an expression of propagating lateral slab tear beneath the Apennines, Basin Research, 24(4), 456-474, https://doi.org/10.1111/j.1365-2117.2011.00534.x.

Beaumont, C., M. H. Nguyen, R. A. Jamieson, and S. Ellis (2006), Crustal flow modes in large hot orogens, Geological Society, London, Special Publications, 268(1), 91-145, doi:10.1144/gsl.sp.2006.268.01.05.

Brune, S., S. E. Williams, N. P. Butterworth, and R. D. Muller (2016), Abrupt plate accelerations shape rifted continental margins, Nature, 536(7615), 201-204, doi:10.1038/nature18319.

de Boorder, H., W. Spakman, S. H. White, and M. J. R. Wortel (1998), Late Cenozoic mineralization, orogenic collapse and slab detachment in the European Alpine Belt, Earth and Planetary Science Letters, 164(3-4), 569-575, doi:10.1016/s0012-821x(98)00247-7.

Osmundsen, P. T., B. Bakke, A. K. Svendby, T. B. Andersen, 2000. "Architecture of the Middle Devonian Kvamshesten Group, western Norway: sedimentary response to deformation above a ramp-flat extensional fault", New Perspectives on the Old Red Sandstone, P. F. Friend, B. P. J. Williams

Stefano Patruno, Henk Kombrink, Stuart G. Archer, 2022. "Cross-border stratigraphy of the Northern, Central and Southern North Sea: a comparative tectono-stratigraphic megasequence synthesis", Cross-Border Themes in Petroleum Geology I: The North Sea, S. Patruno, S. G. Archer, D. Chiarella, J. A. Howell, C. A-L. Jackson, H. Kombrink

Tetreault, J. L., and S. J. H. Buiter (2012), Geodynamic models of terrane accretion: Testing the fate of island arcs, oceanic plateaus, and continental fragments in subduction zones, Journal of Geophysical Research: Solid Earth, 117(B8), doi:10.1029/2012jb009316.

van der Meulen, M.J., J.E. Meulenkamp, and M.J.R. Wortel, Lateral shifts of Apenninic foredeep depocentres reflecting detachment of subducted lithosphere, Earth Planet. Sci. Lett., 154, 203-219, 1998

Wiest, J. D., Wrona, T., Bauck, M. S., Fossen, H., Gawthorpe, R. L., Osmundsen, P. T., & Faleide, J. I. (2020). From Caledonian collapse to North Sea rift: The extended history of a metamorphic core complex. *Tectonics*, 39, https://doi.org/10.1029/2020TC006178